# INSTANCE-DEPENDENT FIXED-BUDGET PURE EXPLORATION IN REINFORCEMENT LEARNING

**Yeongjong Kim**
Department of Mathematics
POSTECH
kim.yj@postech.ac.kr

**Yeoneung Kim**
Department of Applied Artificial Intelligence
Seoultech
kimyeoneung@gmail.com

**Kwang-Sung Jun**
Department of Computer Science & Engineering / Graduate School of AI
POSTECH
kwangsungjun@postech.ac.kr

## ABSTRACT

We study the problem of fixed budget pure exploration in reinforcement learning. The goal is to identify a near-optimal policy, given a fixed budget on the number of interactions with the environment. Unlike the standard PAC setting, we do not require the target error level $\varepsilon$ and failure rate $\delta$ as input. We propose novel algorithms and provide, to the best of our knowledge, the first instance-dependent $\varepsilon$-uniform guarantee, meaning that the probability that $\varepsilon$-correctness is ensured can be obtained simultaneously for all $\varepsilon$ above a budget-dependent threshold. It characterizes the budget requirements in terms of the problem-specific hardness of exploration. As a core component of our analysis, we derive a $\varepsilon$-uniform guarantee for the multiple bandit problem—solving multiple multi-armed bandit instances simultaneously—which may be of independent interest. To enable our analysis, we also develop tools for reward-free exploration under the fixed-budget setting, which we believe will be useful for future work.

## 1 INTRODUCTION

Reinforcement Learning (RL) theory Agarwal et al. (2019) has been studied under two main objectives: regret minimization and policy identification, also known as pure exploration. While the former focuses on maximizing cumulative reward during learning, the latter aims to identify a near-optimal policy without concern for rewards gained during learning. A substantial body of work on policy identification has focused on the fixed-confidence setting Kearns and Singh (2002). This line of research, often referred to as Probably Approximately Correct (PAC) RL, requires the algorithm to spend as many samples as possible until it can find an $\varepsilon$-optimal policy with probability at least $1 - \delta$. Specifically, the algorithm is required to *verify* itself that the returned arm is indeed $\varepsilon$-optimal policy – otherwise, it is not a fixed confidence algorithm. Due to the verification requirement, both $\varepsilon$ and $\delta$ are input to the algorithm. Thus, the analysis must be done for the correctness of the *verification* (i.e., proving that the returned arm is indeed an $\varepsilon$-optimal policy) as well as the sample complexity (i.e., proving how many samples are taken before stopping).

However, the fixed-confidence setting is not the only way to perform policy identification. The fixed-budget setting has been popular in multi-armed bandits (Even-Dar et al., 2006; Bubeck et al., 2009). In this setting, the learner is given a fixed number of interactions with the environment as a budget and is required to output a good policy after exhausting the budget. This setting has numerous merits. First, this setting is arguably more practical because the user of the algorithm can control the budget explicitly. In contrast, the fixed confidence setting assumes that the algorithm can use as many samples as possible (though less is preferred). When stopped forcefully to satisfy practical constraints, it is hard to guarantee the quality of the returned policy. Second, the fixed budget setting has potential to guarantee a better sample complexity because there is no verification requirement (i.e., the algorithm itself certifies that the returned policy is $\varepsilon$-optimal). This was true for multi-armed

bandits where instant-dependent accelerated rates can be obtained as a function of how many good arms there are, and also a data-poor regime guarantee can be obtained, meaning that where a nontrivial performance guarantee is obtained even if the sampling budget is smaller than the number of arms, depending on the problem instance Zhao et al. (2023). These bounds are not likely to be obtained in the fixed confidence setting due to the verification requirement unless extra knowledge about the best arm is known such as Chaudhuri and Kalyanakrishnan (2017). While the $\varepsilon$-correctness verification from the fixed-confidence setting can be necessary in mission-critical applications, there are many applications that do not require such a guarantee, in which case the parameters $\varepsilon$ and $\delta$ becomes a cumbersome hyperparameter.

Despite the desirable properties of the fixed-budget setting in bandit problems, its counterpart in MDPs remains largely unexplored to our knowledge. In this paper, we take the first step at studying fixed-budget policy identification in MDPs, providing new theoretical insights and algorithms that bridge this gap. Specifically, a fixed budget algorithm is required to take in a episode budget $B$ and return a policy $\hat{\pi}$ at the end of $B$-th episode. Our central interest is to upper bound the probability that the algorithm fails to return an $\varepsilon$-optimal policy as an exponentially decaying function of the budget $B$ and instance-dependent quantities, *simultaneously for all $\varepsilon \geq \varepsilon'$* for some budget dependent threshold $\varepsilon'$. We refer to this type of theoretical guarantee as an *$\varepsilon$-uniform guarantee*. In other words, the degree of suboptimality of the learned policy $\hat{\pi}$ is a random variable, and we are characterizing its distribution, in particular its tail behavior.

**Contributions.**   Our main contributions are as follows:

- We propose a novel algorithm, **BREA** (Backward Reachability Estimation and Action elimination), which is, to the best of our knowledge, the first fixed-budget pure exploration algorithm for episodic MDPs with instance-dependent $\varepsilon$-uniform guarantees. The algorithm only requires the episode budget $B$ as an input, and does not assume the uniqueness of the optimal action.

- For the first time, we establish an $\varepsilon$-uniform guarantee for the SAR algorithm (Bubeck et al., 2013) for the muliple bandit problem. This may be of independent interest.

- We develop algorithmic and analytical tools for fixed-budget reward-free exploration by carefully adapting a fixed-confidence reward-free exploration algorithm, L2E (Wagenmaker et al., 2022), to the fixed-budget setting. We prove an $\varepsilon$-uniform guarantee for our fixed-budget reward-free algorithms.

## 2   PRELIMINARIES

**Finite-horizon MDP.**   We consider a finite-horizon non-stationary Markov Decision Process (MDP) defined by the tuple $\mathcal{M} = (\mathcal{S}, \mathcal{A}, H, \{P_h\}_{h=0}^{H-1}, \{R_h\}_{h=1}^{H})$, where $\mathcal{S}$ is a finite set of states of size $S$, $\mathcal{A}$ is a finite set of actions of size $A$, $H \in \mathbb{N}$ is the horizon, $P_0 \in \Delta(\mathcal{S})$ is the initial distribution, $P_h : \mathcal{S} \times \mathcal{A} \to \Delta(\mathcal{S})$ is the transition kernel, and $R_h : \mathcal{S} \times \mathcal{A} \to \Delta([0, 1])$ is the random rewards with $\mathbb{E}[R_h(s,a)] = r_h(s,a)$. $\{P_h\}_{h=0}^{H-1}$ and $\{R_h\}_{h=1}^{H}$ are unknown to the learner.

The initial state $s_1$ is drawn from the initial distribution $P_0$. At each step $h$, taking action $a_h$ in state $s_h$ results in a next state $s_{h+1}$ sampled from the transition kernel $P_h(\cdot \mid s_h, a_h)$. A trajectory $\{(s_h, a_h, R_h(s_h, a_h))\}_{h=1}^{H}$ is called an *episode*, and when the learner reaches the end of the episode, a new episode begins.

A *policy* $\pi = (\pi_1, \ldots, \pi_H)$ is a sequence of decision rules $\pi_h : \mathcal{S} \to \Delta(\mathcal{A})$ for each step $h \in [H]$. The Q-value function of a policy $\pi$ at step $h \in [H]$ is defined as

$$Q_h^\pi(s, a) := \mathbb{E}_\pi\big[\sum_{h'=h}^{H} R_{h'}(s_{h'}, a_{h'}) | s_h = s, a_h = a\big]$$

and it represents the expected reward obtained by choosing action $a$ in state $s$ at step $h$ and choosing the subsequent actions according to the policy $\pi$. The value function of $\pi$ at step $h$ is defined as

$$V_h^\pi(s) = \mathbb{E}_\pi[Q_h^\pi(s, \pi_h(s))]$$

and it represents the expected reward obtained by choosing actions according to the policy $\pi$ starting in state $s$ at step $h$. We also define $V_0^\pi := \mathbb{E}_{s \sim P_0}[V_1^\pi(s)]$. The optimal Q-value function, optimal value function are defined as

$$Q_h^*(s,a) = \sup_\pi Q_h^\pi(s,a), \quad V_h^*(s) = \sup_\pi V_h^\pi(s), \quad V_0^* = \sup_\pi V_0^\pi.$$

Throughout the paper, we do not assume that the optimal action or policy is unique.

**Pure exploration under the fixed budget setting.** In pure exploration under the fixed budget setting, the goal is to identify an optimal policy $\pi^*$ (or near-optimal) based on a limited interaction budget. Specifically, the learner is allowed to execute a total of $B$ episodes and must return a single policy $\hat{\pi}$ at the end. The performance is measured by the simple regret, which is defined as

$$V_0^* - V_0^{\hat{\pi}}.$$

A policy $\hat{\pi}$ is called $\varepsilon$-*good* if $V_0^* - V_0^{\hat{\pi}} \le \varepsilon$. In this paper, we propose an algorithm and prove their performance guarantee by showing some instance-dependent upper bounds of the failure probability

$$\mathbb{P}(V_0^* - V_0^{\hat{\pi}} > \varepsilon).$$

**Instance-dependent quantities.** To capture the instance-dependent complexity of the problem, we need the notion of *suboptimality gaps* defined as

$$\Delta_h(s,a) := V_h^*(s) - Q_h^*(s,a),$$
$$\Delta_h^\pi(s,a) := \max_{a'} Q_h^\pi(s,a') - Q_h^\pi(s,a).$$

For our analysis, we also denote

$$\bar{\Delta}_h(s,a) := \begin{cases} \Delta_h(s,a), & \text{if } Q_h^*(s,a) < V_h^*(s) \\ \Delta_h(s,a'), & \text{if } Q_h^*(s,a) = V_h^* \text{ and } a' \text{ is the second best action,} \end{cases}$$

$$\bar{\Delta}_h^\pi(s,a) := \begin{cases} \Delta_h^\pi(s,a), & \text{if } Q_h^\pi(s,a) < \max_{a'} Q_h^\pi(s,a') \\ \Delta_h^\pi(s,\tilde{a}), & \text{if } Q_h^\pi(s,a) = \max_{a'} Q_h^\pi(s,a') \text{ and } \tilde{a} \text{ is the second best action with respect to } \pi. \end{cases}$$

Thus, if the optimal action in $s \in \mathcal{S}$ at step $h$ is unique, $\bar{\Delta}_h(s,a) > 0$ for all $a \in \mathcal{A}$. In contrast, if there are multiple optimal actions in $s \in \mathcal{S}$ at step $h$, $\bar{\Delta}_h(s,a) = 0$ for all optimal actions $a$. Similar results hold for $\bar{\Delta}_h(s,a)$ as well.

In MDP, the probability of reaching each state or action is important. Let $\pi$ be a policy, $s \in \mathcal{S}, a \in \mathcal{A}, h \in [H], \mathcal{Z} \subset \mathcal{S} \times \mathcal{A}$, we use the following notations:

$$w_h^\pi(s) = \mathbb{P}_\pi[s_h = s], \qquad w_h^\pi(s,a) = \mathbb{P}_\pi[s_h = s, a_h = a], \qquad w_h^\pi(\mathcal{Z}) = \mathbb{P}_\pi[(s_h, a_h) \in \mathcal{Z}],$$
$$W_h(s) = \sup_\pi w_h^\pi(s) = \sup_\pi w_h^\pi(s,a), \qquad W_h(\mathcal{Z}) = \sup_\pi w_h^\pi(\mathcal{Z}).$$

We refer to $w_h^\pi(\cdot)$ as the *occupancy measure* and $W_h(\cdot)$ as the *reachability*. Using these notions, we define the *controllability* of MDP at step $h$ as

$$C_h := \sup_\pi \sum_{s, W_h(s) > 0} \frac{w_h^\pi(s)}{W_h(s)}.$$

Then, we have

$$1 = \sup_\pi \sum_{s, W_h(s) > 0} w_h^\pi(s) \le C_h = \sup_\pi \sum_{s, W_h(s) > 0} \frac{w_h^\pi(s)}{W_h(s)} \le \sum_{s, W_h(s) > 0} \sup_\pi \frac{w_h^\pi(s)}{W_h(s)} \le S.$$

We can see that $C_h = 1$ if $W_h(s) = 0$ or $1$ for any state $s$ i.e. the learner can reach $s_h = s$ with probability 1 by some policy for any reachable state $s$. On the other hand, $C_h = S$ if $w_h^\pi(s) = W_h(s) > 0$ for any state $s \in \mathcal{S}$, any policy $\pi$ i.e. the learner cannot control the occupancy measure by varying policy and all states are reachable. Therefore, intuitively, a larger $C_h$ indicates that the MDP is more difficult to control at step $h$.

## 3 THE BREA ALGORITHM

There are inherent difficulties in achieving instance-dependent $\varepsilon$-uniform guarantee for fixed budget setting. First, while it is relatively straightforward to analyze algorithms in the fixed confidence setting using concentration bounds such as Hoeffding or Bernstein bound with a prespecified confidence level $\delta$, it is much more challenging in the fixed budget setting, where neither the confidence level $\delta$ nor the accuracy level $\varepsilon$ is known in advance. Second, whereas the fixed-confidence setting typically allows for a potentially excessive number of samples before termination (depending on the confidence level), the fixed-budget setting strictly limits the algorithm to a finite number of samples. Third, it is hard to simply convert the fixed-confidence algorithms since it requires the knowledge of nontrivial instance-dependent terms. Even if it is possible, the conversion of fixed-confidence algorithm would require not only the budget $B$ but also one of the confidence $\delta$ and the accuracy $\varepsilon$. Also, theoretical guarantee of this conversion would only applies to prespecified $\varepsilon$ (or $\delta$), which is much weaker than $\varepsilon$-uniform guarantee. In this section, we present how we design and analyze our algorithm to overcome the aforementioned difficulties.

At step $h$, each state $s$ can be treated as a bandit problem, where the expected reward of each action $a$ is given by $Q_h^*(s, a)$. If we aim to learn the exact optimal policy maximizing $Q_h^*(s, a)$, we need to sample trajectories $s_{h+1}, a_{h+1}, \ldots, s_H, a_H$ generated under the optimal policy $\{\pi_{h'}^*\}_{h'=h+1}^H$, which is unknown. Fortunately, since our goal is to learn an approximately optimal policy, the following proposition shows that it suffices to use a suitably accurate policy $\{\hat{\pi}_{h'}\}_{h'=h+1}^H$ for sampling in order to learn $\hat{\pi}_h$.

**Proposition 1.** *([Wagenmaker et al., 2022, Lemma B.1) Assume that some deterministic policy $\hat{\pi}$ satisfies $\Delta_h^{\hat{\pi}}(s, \hat{\pi}_h(s)) \leq \varepsilon_h(s)$ for any $h' \leq h \leq H$ and any $s \in \mathcal{S}$. Then, for any policy $\pi'$,*

$$\sum_s w_{h'}^{\pi'}(s) \left( V_{h'}^*(s) - V_{h'}^{\hat{\pi}}(s) \right) \leq \sum_{h=h'}^H \sup_\pi \sum_s w_h^\pi(s) \varepsilon_h(s).$$

Note that $\Delta_h^{\hat{\pi}}(s, a)$ depends only on the future policies $\{\hat{\pi}_{h'}\}_{h'=h+1}^H$, implying that we must determine them before learning $\hat{\pi}_h(s)$. By this observation, our learning proceeds backward from $H$ to $1$.

If we assume that the hypothesis of the previous proposition holds with $h' = 1$ and $\varepsilon_h(s) := \frac{\varepsilon}{C_h H W_h(s)}$, then the proposition says

$$
\begin{aligned}
V_0^* - V_0^{\hat{\pi}} &\leq \sum_{h=1}^H \sup_\pi \sum_s w_h^\pi(s) \varepsilon_h(s) \\
&= \sum_{h=1}^H \sup_\pi \sum_s w_h^\pi(s) \frac{\varepsilon}{C_h H W_h(s)} \\
&= \sum_{h=1}^H \frac{\varepsilon}{H} && \text{(definition of } C_h) \\
&= \varepsilon.
\end{aligned}
$$

Therefore, we design our algorithm to identify a $\Theta(\frac{\varepsilon}{C_h H W_h(s)})$-good action for each relevant state $s$. The precise definition of "relevant state" will be given in the analysis. We again emphasize that $\varepsilon$ is not an input to our algorithm and can be chosen arbitrarily for the purpose of analysis. Our algorithm consists of two key components: estimating the reachability $W_h(s)$ and eliminating actions. We introduce the following notation, which will be used in the statements of upcoming results.

$$\varepsilon_B := (1 + \frac{\log(2)B}{c(B)})^{-0.6321}$$

denotes an error threshold that depends on the budget $B$. The factor

$$C_{\text{L2E}}(B) = \tilde{O}(\text{poly}(S, A, H)),$$

is formally defined in Appendix C, equation 5. We denote $C_{\text{L2E}}(B) = SH^2 c(B)$.

---

**Algorithm 1** **F**ixed **B**udget **L**earn to **E**xplore (FB-L2E)

---

 1: **function** FB-L2E($\mathcal{X} \subseteq \mathcal{S} \times \mathcal{A}$, step $h$, budget $B$)
 2:     **if** $|\mathcal{X}| = 0$ **then**
 3:         **return** $\{(\emptyset, \emptyset, 0)\}$
 4:     **end if**
 5:     $J \leftarrow \lceil 0.6321 \log_2(1 + \frac{\log(2)B}{c(B)}) \rceil$ ($c(B)$ is defined in Appendix C)
 6:     **for** $j = 1$ to $J$ **do**
 7:         $L_j \leftarrow 2^{J-j}, \quad \delta_j \leftarrow (\frac{1}{8SAH})^{0.6321 L_j \log\log(8SAH)}$
 8:         $K_j \leftarrow K_j(\delta_j, SAH\delta_j)$ ($K_j$ is defined in Appendix C)
 9:         $N_j \leftarrow K_j/(4|\mathcal{X}| \cdot 2^j)$
10:         $(\mathcal{X}_j, \Pi_j) \leftarrow \text{FINDEXPLORABLESETS}(\mathcal{X}, h, \delta, K_j, N_j)$
11:         $\mathcal{X} \leftarrow \mathcal{X} \setminus \mathcal{X}_j$
12:     **end for**
13:     **return** $\{(\mathcal{X}_j, \Pi_j, N_j)\}_{j=1}^J$
14: **end function**
15:
16: **function** FINDEXPLORABLESETS($\mathcal{X} \subseteq \mathcal{S} \times \mathcal{A}$, step $h$, confidence $\delta$, epochs $K$, samples $N$)
17:     $r_h^1(s, a) \leftarrow 1$ if $(s, a) \in \mathcal{X}$, else 0
18:     $N(s, a, h) \leftarrow 0, \mathcal{Y} \leftarrow \emptyset, \Pi \leftarrow \emptyset, j \leftarrow 1$
19:     **for** $k = 1$ to $K$ **do**
20:         // StrongEuler is as defined in Simchowitz and Jamieson (2019)
21:         Run STRONGEULER($\delta$) on reward $r_h^j$ to get trajectory $\{(s_h^k, a_h^k, h)\}_{h=1}^H$ and policy $\pi_k$
22:         $N(s_h^k, a_h^k) \leftarrow N(s_h^k, a_h^k) + 1, \quad \Pi \leftarrow \Pi \cup \{\pi_k\}$
23:         **if** $N(s_h^k, a_h^k) \geq N$, $(s_h^k, a_h^k) \in \mathcal{X}$ and $(s_h^k, a_h^k) \notin \mathcal{Y}$ **then**
24:             $\mathcal{Y} \leftarrow \mathcal{Y} \cup (s_h^k, a_h^k)$
25:             $r_h^{j+1}(s, a) \leftarrow 1$ if $(s, a) \in \mathcal{X} \setminus \mathcal{Y}$, else 0
26:             $j \leftarrow j + 1$
27:             Restart STRONGEULER($\delta$)
28:         **end if**
29:     **end for**
30:     **return** $\mathcal{Y}, \Pi$
31: **end function**

---

### 3.1 REACHABILITY ESTIMATION

The first part of our algorithm is greatly influenced by Wagenmaker et al. (2022). Through the first part, we estimate the reachability $W_h(s)$ of each state $s$ at step $h$. To this end, we execute a fixed-budget reward-free exploration. One notable benefit of reward-free exploration is that it only needs to be run once, after which the collected data can be applied to a variety of downstream reward functions. More specifically, we reset the reward as $R_{h'}(s', a') = \begin{cases} 1, & \text{if } (s', a', h') = (s, 1, h), \\ 0, & \text{otherwise.} \end{cases}$,

where we arbitrarily fix an action and denote it by 1. With this reset reward, an optimal policy maximizes the visitation probability of $(s, 1)$ at step $h$. Therefore, $V_0^* = W_h(s, 1) = W_h(s)$. To approximate such an optimal policy, we employ STRONGEULER (Simchowitz and Jamieson, 2019).

More generally, the reachability $W_h(\mathcal{X})$ of any subset $\mathcal{X} \subset \mathcal{S} \times \mathcal{A}$ can be estimated in the same manner. We formalize this in Algorithm 1, which we refer to as **FB-L2E**, short for *Fixed-Budget Learn2Explore*. It is a careful adaptation of Learn2Explore algorithm introduced in Wagenmaker et al. (2022), which itself is inspired by Zhang et al. (2021); Brafman and Tennenholtz (2003).

Algorithm 1 satisfies the following guarantee:

**Theorem 3.1.** *Consider running Algorithm 1 with $B \geq c(B)$. Then, the following statements hold.*

    *1. The total budget used is at most $B$.*

    *2. For any $\varepsilon \geq 2SH^2\varepsilon_B$, with probability at least $1 - \exp\left(-\tilde{\Theta}\left(\frac{\varepsilon B}{C_{\text{L2E}}(B)}\right)\right)$,*

*(1) The reachability of each set $\mathcal{X}_i$ satisfies*

$$\frac{|\mathcal{X}_i|}{|\mathcal{X}|} \cdot 2^{-i-3} \leq W_h(\mathcal{X}_i) \leq 2^{-i+1} \quad \text{for all } i \leq i_\varepsilon := \left\lceil \log_2\left(\frac{2SH^2}{\varepsilon}\right) \right\rceil,$$

*(2) The remaining elements, $\bar{\mathcal{X}} := \mathcal{X} \setminus \cup_{i=1}^{i_\varepsilon} \mathcal{X}_i$ satisfy*

$$\sup_\pi \sum_{(s,a) \in \bar{\mathcal{X}}} w_h^\pi(s,a) \leq \frac{\varepsilon}{2SH^2}.$$

*(3) Moreover, for any $i \leq i_\varepsilon$, if each policy in $\Pi_i$ is executed $A$ times, then every state-action pair $(s,a) \in \mathcal{X}_i$ is visited at least $\frac{1}{8}AN_i$ times.*

*Here, the probability accounts for both the randomness in execution and resampling.*

The proof of Theorem 3.1 is deferred to Appendix C.

**Remark 2.** *Theorem 3.1 crucially relies on the fact that* STRONGEULER *(Simchowitz and Jamieson, 2019) achieves a high probability regret bound with $\log \frac{1}{\delta}$ dependence. However, when the target set is $\mathcal{X} = \{(s,a)\}$, similar results can be obtained by applying a boosting technique even if we use other algorithms with worse dependence. Although we only present Algorithm 1 in the main text for the simplicity, the algorithm with boosting technique is described in Appendix C, Algorithm 0.*

### 3.2 ACTION ELIMINATION

In the second part of our algorithm, we iteratively sample trajectories, compute empirical Q-function of state-action pairs, and eliminate suboptimal actions. For the purpose of efficient elimination, we employ a multiple bandit algorithm, *Successive Accepts and Rejects* (**SAR**), proposed by Bubeck et al. (2013), and, for the first time, provide an $\varepsilon$-correctness guarantee for this algorithm. By employing this algorithm to our main algorithm, we are able to reduce the dependency on $S$ compared to applying its multi-armed bandit counterpart. For a more detailed explanation, see Appendix D, Remark 27.

**Multiple bandit problem.** Consider $M$ instances of multi-armed bandit problems, each with $K$ arms. Each arm $i$ in instance $m$ yields stochastic rewards supported on $[0, \sigma]$, with mean $\mu_{m,i}$, ordered such that $\mu_{m,1} \geq \cdots \geq \mu_{m,K}$. We denote each bandit-arm pair by $(m, i)$, where $m \in [M]$ and $i \in [K]$. The objective is to identify a good arm in each instance $m \in [M]$ under a total budget of $B$ pulls.

We now define some notations. Let $\widehat{\mu}_{m,i}(n)$ denote the empirical mean reward of arm $i$ in instance $m$ after $n$ pulls. Define the suboptimality gap as

$$\bar{\Delta}_{m,i} := \begin{cases} \mu_{m,1} - \mu_{m,2}, & \text{if } i = 1, \\ \mu_{m,1} - \mu_{m,i}, & \text{if } i \in \{2, \ldots, K\}. \end{cases}$$

We enumerate all gaps $\bar{\Delta}_{m,i}$ over all $(m, i) \in [M] \times [K]$ in increasing order as

$$\bar{\Delta}_{(1)} \leq \bar{\Delta}_{(2)} \leq \cdots \leq \bar{\Delta}_{(MK)}.$$

Let

$$g(\varepsilon) := \left| \left\{ (m, i) \in [M] \times [K] : \mu_{m,1} - \mu_{m,i} \leq \varepsilon \right\} \right|$$

for any $\varepsilon > 0$, and define the harmonic log term

$$\overline{\log}(MK) := \frac{1}{2} + \sum_{i=2}^{MK} \frac{1}{i}.$$

For each $k \in [MK - 1]$, define

$$n_k(B, M, K) := \left\lceil \frac{1}{\overline{\log}(MK)} \cdot \frac{B - MK}{MK + 1 - k} \right\rceil. \tag{1}$$

The SAR algorithm (Bubeck et al., 2013) is summarized in Algorithm 2. By leveraging the ranking of empirical gaps, SAR adaptively distributes the budget across bandit instances, solving the multiple bandit problem efficiently. We present a theoretical guarantee for its ability to identify $\varepsilon$-good arms.

---

**Algorithm 2** **S**uccessive **A**ccept and **R**eject (SAR) for the multiple bandit

1: **input:** Budget $B$
2: $A_1 \leftarrow \{(1,1), \ldots, (M,K)\}, n_0 \leftarrow 0$
3: **for** $k = 1$ to $MK - 1$ **do**
4:     $n_k \leftarrow n_k(B, M, K)$ (as defined in equation 1)
5:     $\forall (m,i) \in A_k,$    pull $(m,i)$ for $n_k - n_{k-1}$ times
6:     $\forall m,$    $\hat{1}_m \leftarrow \arg\max_{i:(m,i)\in A_k} \hat{\mu}_{m,i}(n_k)$ (Break ties arbitrarily)
7:     **if** $\exists m$ such that $\hat{1}_m$ is the last active arm in $m$ **then**
8:         $J_m = \hat{1}_m$ (Accept)
9:         $A_{k+1} \leftarrow A_k \setminus \{(m, \hat{1}_m)\}$ (Deactivate)
10:    **else**
11:        $(m_k, i_k) \leftarrow \arg\max_{(m,i)\in A_k} \left( \hat{\mu}_{m,\hat{1}_m}(n_k) - \hat{\mu}_{m,i}(n_k) \right)$ (Break ties arbitrarily)
12:        $A_{k+1} \leftarrow A_k \setminus \{(m_k, i_k)\}$ (Reject and deactivate)
13:    **end if**
14: **end for**
15: $J_m \leftarrow i$ for $A_{MK} = \{(m,i)\}$
16: **return** $\{(m, J_m)\}_{m=1}^M$

---

**Theorem 3.2.** *If we run Algorithm 2 with $B \geq MK$, then the total number of budget used is at most $B$ and*

$$\mathbb{P}(\exists m \in [M] : \mu_{m,1} - \mu_{m,J(m)} > \varepsilon) \leq 2M^2 K^2 \exp\left(-\frac{B - MK}{128\sigma^2 \overline{\log}(MK) \cdot \sum_{i\in[MK]} (\bar{\Delta}_{(i)} \vee \varepsilon)^{-2}}\right).$$

*for any $\varepsilon \geq 0$.*

The proof of Theorem 3.2 is deferred to Appendix D.

### 3.3 OVERVIEW OF THE BREA ALGORITHM

We combine the two mechanisms described above to construct our main algorithm. The algorithm proceeds in a backward manner over steps $h = H, H - 1, \ldots, 1$. At each step $h$, the first half of the budget is devoted to estimating the reachability $W_h(s)$ for each state $s$, while the second half applies the SAR mechanism to eliminate suboptimal actions. Although the logic by which our algorithm eliminates actions is entirely different, the structure of eliminating actions after reward-free exploration was also used in the fixed-confidence algorithm, MOCA (Wagenmaker et al., 2022).

In general MDPs, the stochasticity of the transition kernel prevents us from freely collecting arbitrary state-action samples. However, Theorem 3.1 ensures that, with high probability, the policies stored during the reachability estimation phase yield sufficient samples for each relevant state-action pair. Under this event, the SAR mechanism is expected to perform reliably. We now present our main theorem and its corollary; their proofs are provided in Appendix E.

**Theorem 3.3.** *If we run Algorithm 3 with*

$$B \geq \max\{2SHc(\frac{B}{2SH}), 2SA\varepsilon_{\frac{B}{2SH}} \log_2 \frac{1}{\varepsilon_{\frac{B}{2SH}}}\},$$

*then the total number of budget used is at most $B$. Moreover, for any $\varepsilon \geq 2SH^2\varepsilon_{\frac{B}{2SH}}$,*

$$\mathbb{P}\left(V_0^* - V_0^{\hat{\pi}} > \varepsilon\right) \leq \exp\left(-\tilde{\Theta}\left(\frac{\varepsilon B}{C_{\text{L2E}}(\frac{B}{2SH})}\right)\right)$$

$$+ \exp\left(-\tilde{\Theta}\left(\frac{B}{H^5 \max_{h\in[H]} C_h^2 \sum_{s\in\mathcal{S}} W_h(s)^{-1} \sum_{a\in\mathcal{A}} (\bar{\Delta}_h(s,a) \vee \frac{\varepsilon}{W_h(s)})^{-2}}\right)\right).$$

**Corollary 3.** *In addition to the hypothesis of Theorem 3.3, assume further that $2SH^2\varepsilon_{\frac{B}{2SH}} < \varepsilon^* :=$ $\min\{\min_{s,h}^+ W_h(s), 2H \min_{s,a,h}^+ C_h W_h(s)\bar{\Delta}_h(s,a)\}$ and the optimal action in each state $s$ at each*

---

**Algorithm 3** **B**ackward **R**eachability **E**stimation and **A**ction elimination (BREA)

1: **input:** Budget $B$
2: $B' \leftarrow \lfloor \frac{B}{2SH} \rfloor, \quad J \leftarrow \lceil 0.6321 \log_2(1 + \frac{\log(2)B'}{c(B')}) \rceil$
3: $B'' \leftarrow \frac{B}{2HJ}$
4: **for** $h = H, H-1, \ldots, 1$ **do**
5: $\quad \mathcal{Z}_h \leftarrow \emptyset$
6: $\quad$ **for** $s \in \mathcal{S}$ **do** $\{(\mathcal{X}_j^{sh}, \Pi_j^{sh}, N_j^{sh})\}_{j=1}^J \leftarrow$ FB-L2E$(\{(s,1)\}, h, B')$ (1 is an arbitrary action)
7: $\quad\quad$ **if** $\mathcal{X}_h^{sh} = \{(s,1)\}$ for some $j \in [J]$ **then**
8: $\quad\quad\quad \widehat{W}_h(s) \leftarrow 2^{-j+1}, \quad \mathcal{Z}_h \leftarrow \mathcal{Z}_h \cup \{s\}$
9: $\quad\quad$ **end if**
10: $\quad$ **end for**
11: $\quad$ **for** $i = 1$ to $J$ **do**
12: $\quad\quad \mathcal{Z}_{hi} \leftarrow \{s \in \mathcal{Z}_h : \widehat{W}_h(s) = 2^{-i+1}\}, \quad A_1 \leftarrow \mathcal{Z}_{hi} \times \mathcal{A},$
13: $\quad\quad \forall(s,a) \in A_1, \quad N(s,a) \leftarrow 0, \quad T(s,a) \leftarrow 0, \quad T_0(s,a) = 0, \quad Q(s,a) \leftarrow 0$
14: $\quad\quad$ **for** $k = 1$ to $|\mathcal{Z}_{hi}|A - 1$ **do**
15: $\quad\quad\quad n_k \leftarrow n_k(\lfloor B'' 2^{-i-2} \rfloor, |\mathcal{Z}_{hi}|, A)$ (as defined in equation 1)
16: $\quad\quad\quad$ **for** $(s,a) \in A_k$ **do**
17: $\quad\quad\quad\quad T_k(s,a) \leftarrow \lfloor \frac{n_k}{N_i^{sh}} \rfloor$
18: $\quad\quad\quad\quad$ Rerun each policy in $\Pi_i^{sh}$ for $T_k(s,a) - T_{k-1}(s,a)$ times
19: $\quad\quad\quad\quad$ **for** each time $t = T(s,a) + 1$ to $T_k(s,a)$ **do**
20: $\quad\quad\quad\quad\quad$ **if** $(s,a)$ is visited at step $h$ **then**
21: $\quad\quad\quad\quad\quad\quad$ Take action $a$ and extend a trajectory using $\{\hat{\pi}_{h'}\}_{h'=h+1}^H$
22: $\quad\quad\quad\quad\quad\quad N(s,a) \leftarrow N(s,a) + 1$
23: $\quad\quad\quad\quad\quad\quad Q(s,a) \leftarrow Q(s,a) + \sum_{h'=h}^H R_{h'}^t(s_{h'}^t, a_{h'}^t)$
24: $\quad\quad\quad\quad\quad$ **end if**
25: $\quad\quad\quad\quad$ **end for**
26: $\quad\quad\quad\quad \hat{Q}_h^{\hat{\pi}}(s,a) \leftarrow Q(s,a)/N(s,a)$ **if** $N(s,a) > 0$ **else** $0$
27: $\quad\quad\quad\quad T(s,a) \leftarrow T(s,a) + T_k(s,a)$
28: $\quad\quad\quad$ **end for**
29: $\quad\quad\quad$ **if** $\exists$ state $s$ with unique surviving pair $(s,a)$ in $A_k$ **then**
30: $\quad\quad\quad\quad \hat{\pi}_h(s) \leftarrow a, \quad A_{k+1} \leftarrow A_k \setminus \{(s,a)\}$
31: $\quad\quad\quad$ **else**
32: $\quad\quad\quad\quad \forall(s,a) \in A_k, \quad \widehat{\Delta}_h^{\hat{\pi}}(s,a) \leftarrow \max_{a:(s,a) \in A_k} \hat{Q}_h^{\hat{\pi}}(s,a) - \hat{Q}_h^{\hat{\pi}}(s,a)$
33: $\quad\quad\quad\quad (s',a') \leftarrow \arg\max_{(s,a) \in A_k} \widehat{\Delta}_h^{\hat{\pi}}(s,a)$ (Break ties arbitrarily)
34: $\quad\quad\quad\quad A_{k+1} \leftarrow A_k \setminus \{(s',a')\}$
35: $\quad\quad\quad$ **end if**
36: $\quad\quad$ **end for**
37: $\quad\quad \hat{\pi}(s) \leftarrow a$ for $A_{|\mathcal{Z}_{hi}|A} = \{(s,a)\}$
38: $\quad$ **end for**
39: $\quad$ For each $s \in \mathcal{S} \setminus \mathcal{Z}_h$, set $\hat{\pi}_h(s)$ as any action
40: **end for**
41: **return** $\hat{\pi}$

---

*step $h$ is unique. Then, we obtain a guarantee of the best policy identification, given by*

$$\mathbb{P}\left(V_0^* - V_0^{\hat{\pi}} > 0\right) \leq \exp\left(-\tilde{\Theta}\left(\frac{\varepsilon^* B}{C_{\text{L2E}}(\frac{B}{2SH})}\right)\right)$$
$$+ \exp\left(-\tilde{\Theta}\left(\frac{B}{H^3 \max_{h \in [H]} \sum_{s \in \mathcal{S}} W_h(s)^{-1} \sum_{a \in \mathcal{A}} \bar{\Delta}_h(s,a)^{-2}}\right)\right).$$

**Remark 4.** *From Theorem 3.3, we can derive the sample complexity required by BREA to identify an $\varepsilon$-correct policy with probability at least $1 - \delta$, given by*

$$\tau_{\varepsilon,\delta} = \tilde{\Theta}\left(\frac{\text{poly}(S,A,H)}{\varepsilon} + H^5 \max_{h \in [H]} C_h^2 \sum_{s \in \mathcal{S}} \frac{1}{W_h(s)} \sum_{a \in \mathcal{A}} \frac{1}{\left(\bar{\Delta}_h(s,a) \vee \frac{\varepsilon}{W_h(s)}\right)^2}\right) \log \frac{1}{\delta}.$$

*The first term inside $\tilde{\Theta}$ is a lower-order term. The second term inside $\tilde{\Theta}$ becomes $\sum_{a \in \mathcal{A}} \frac{1}{(\Delta(a) \vee \varepsilon)^2}$ for multi-armed bandits ($S = H = 1$). This is consistent with known results in the bandit literature ((Even-Dar et al., 2006; Audibert et al., 2010; Karnin et al., 2013)). It is also noteworthy that our sample complexity is deterministic while the sample complexity of PAC RL algorithm typically is guaranteed with probability at least $1 - \delta$.*

**Remark 5.** *Our sample complexity involves $H^5 \max_h$ term, in contrast to the $H^4 \sum_h$ dependence that appear in PAC RL literature ((Wagenmaker et al., 2022; Wagenmaker and Jamieson, 2022; Tirinzoni et al., 2023)). This difference stems from the inherent difficulty of the fixed budget setting, where the algorithm does not know in advance how to distribute the budget across different $h$. A similar issue regarding the dependency on $S$ could be resolved by employing a multiple bandit algorithm instead of a multi-armed bandit algorithm.*

## 3.4 INCORPORATING TARGET ACCURACY

If an accuracy level $\varepsilon$ is provided as input, we can modify Algorithm 3 to include a third part and obtain a different form of probabilistic guarantee. While Algorithm 3 allocates $\frac{B}{4}$ budget to each of its two parts, the modified algorithm assigns $\frac{B}{4}$ to the first and the second part and assgins $\frac{B}{2}$ to the last part.

In the third part, for each multiple-bandit instance $\mathcal{Z}_{hi}$, let $\hat{g}_{hi}^{\hat{\pi}}(\varepsilon)$ denote the number of pairs $(s, a) \in \mathcal{Z}_{hi}$ such that $\widehat{\Delta}_h^{\hat{\pi}}(s, a) \le \frac{\varepsilon}{\widehat{W}_h(s)}$. After the second part, we gather the last $\hat{g}_{hi}^{\hat{\pi}}(\varepsilon)$ surviving pairs and perform an additional refinement step.

Theoretical guarantees for this variant are presented in the next theorem. The full algorithm (Algorithm 5) and its analysis are provided in Appendix F.

**Theorem 3.4.** *(Informal) There exists a variant of Algorithm 3 that, when given a sufficiently large budget $B$ and an accuracy level $\varepsilon \ge 2SH^2 \varepsilon_{\frac{B}{2SH}}$ as input, it uses at most budget $B$ and satisfies the following:*

$$\mathbb{P}\left(V_0^* - V_0^{\hat{\pi}} > \varepsilon\right) \le \exp\left(-\tilde{\Theta}\left(\frac{\varepsilon B}{\text{poly}(S, A, H, \log B)}\right)\right)$$

$$+ \exp\left(-\tilde{\Theta}\left(\frac{B}{H^3 \max_{h \in [H]} \sum_{s \in \mathcal{S}} W_h(s)^{-1} \sum_{a \in \mathcal{A}} (\bar{\Delta}_h(s, a) \vee \frac{\varepsilon}{W_h(s)})^{-2}}\right)\right)$$

$$+ \exp\left(-\tilde{\Theta}\left(\frac{\varepsilon^2 B}{H^5 \max_{h \in [H]} |\text{OPT}_h(3\varepsilon)|}\right)\right),$$

*where $\text{OPT}_h(\varepsilon) = \{(s, a) \in \mathcal{S} \times \mathcal{A} : \Delta_h(s, a) W_h(s) \le \varepsilon\}$.*

**Remark 6.** *From Theorem 3.4, we can derive the sample complexity required by the modified algorithm to identify an $\varepsilon$-correct policy with probability at least $1 - \delta$, given by*

$$\tau_{\varepsilon, \delta} = \tilde{\Theta}\left(\frac{\text{poly}(S, A, H)}{\varepsilon} + H^3 \max_{h \in [H]} \sum_{s \in \mathcal{S}} \frac{1}{W_h(s)} \sum_{a \in \mathcal{A}} \frac{1}{\left(\bar{\Delta}_h(s, a) \vee \frac{\varepsilon}{W_h(s)}\right)^2} + \frac{H^5 \max_{h \in [H]} |\text{OPT}_h(3\varepsilon)|}{\varepsilon^2}\right) \log \frac{1}{\delta}.$$

*It is interesting that even though the logic of action elimination is very different, this expression is closely aligned with the sample complexity*

$$\tau_{\varepsilon, \delta} = \frac{C_{\text{LOT}}(\varepsilon)}{\varepsilon} + \tilde{\Theta}\left(H^2 \sum_{h \in [H]} \sum_{s \in \mathcal{S}} \frac{1}{W_h(s)} \sum_{a \in \mathcal{A}} \frac{1}{\left(\bar{\Delta}_h(s, a) \vee \frac{\varepsilon}{W_h(s)}\right)^2} + \frac{H^4 \sum_{h \in [H]} |\text{OPT}_h(\varepsilon)|}{\varepsilon^2}\right) \log \frac{1}{\delta}$$

*of MOCA algorithm (Wagenmaker et al., 2022), where $C_{\text{LOT}} = \text{poly}(S, A, H, \log \frac{1}{\varepsilon}, \log \frac{1}{\delta})$.*

## 4    CONCLUSION

In this paper, we have explored the fixed-budget setting of the pure exploration MDP, which is surprisingly underexplored in RL theory. While our results establish the first fully instance-dependent guarantee in the fixed budget setting, these are just beginning. First, it would be great to see what kind of instance-dependent acceleration can be proven in MDP, which should be possible given that accelerated rates were possible in bandits as a function of the number of good arms Katz-Samuels and Jamieson (2020); Zhao et al. (2023). Second, similarly, it would be interesting to explore what kind of data-poor regime guarantees are attainable – again, such bounds are available in the bandit setting Katz-Samuels and Jamieson (2020); Zhao et al. (2023). Third, we believe the factor $H^2$ in the sample complexity may be improved by leveraging variance-dependent concentration bounds. Finally, it would be interesting to extend our setting to the function approximation setting.

## REPRODUCIBILITY STATEMENT

We have carefully specified all details of the algorithms presented in this paper. Moreover, we clearly state all assumptions required for the theoretical guarantees of our methods. We believe that this level of detail ensures the reproducibility of our results.

## ACKNOWLEDGEMENTS

This work was supported in part by the National Research Foundation of Korea (NRF) grant funded by the Korea government (MSIT) (RS-2023-00219980), and in part by the Institute of Information & Communications Technology Planning & Evaluation (IITP) grant funded by the Korea government (MSIT) (No. RS-2019-II191906, Artificial Intelligence Graduate School Program (POSTECH)). Kwang-Sung Jun was supported in part by the National Science Foundation under grant CCF-2327013 and Meta Platforms, Inc.

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
