**Notation.** For a positive integer $n$, we write $[n] := \{1, 2, \ldots, n\}$. We use $f = \tilde{\Theta}(g)$ to denote that the ratio $\frac{f}{g}$ is bounded both above and below by polylogarithmic functions. We define $\min^+_{x \in X} f(x) := \min_{x \in X : f(x) > 0} f(x)$. We use poly$(\cdot)$ to denote a polynomial in the variables inside the parentheses. We write $\log$ for natural logarithm and $\log_2$ for binary logarithm.

## A    RELATED WORK

Given the breadth of the literature on each topic, we focus on introducing only the most recent and relevant works.

**Instance-dependent regret minimization in episodic MDPs.** Zanette and Brunskill (2019) proposed the EULER algorithm and proved a regret bound of $\sqrt{SAK \min\{\mathbb{Q}_\star H, \mathcal{G}^2\}}$, where $\mathbb{Q}_\star, \mathcal{G}$ are instance dependent term. Soon after, Simchowitz and Jamieson (2019) proposed STRONGEULER algorithm and proved a gap-dependent regret bound for episodic tabular MDPs, showing that optimistic algorithms can achieve $O\big(\sum_{s,a,h} \frac{\log T}{\Delta_h(s,a)}\big)$ regret. This result, obtained via a novel "clipped" regret decomposition, smoothly interpolates between instance-dependent $O(\log T)$ growth and the worst-case $O(\sqrt{T})$ rate, without requiring simplifying assumptions like a bounded mixing time. Dann et al. (2021) further refined these bounds by defining value-function gaps that ignore states never visited by an optimal policy. Finally, we note that any low-regret algorithm can be converted into a high-probability guarantee on near-optimal performance via an online-to-batch conversion. For detailed explanations, see Jin et al. (2018). However, recent studies ((Wagenmaker et al., 2022; Tirinzoni et al., 2023)) suggest that algorithms for minimizing regret cannot be instance-optimal for identifying good policies, motivating specialized algorithms that explore more strategically than standard optimism.

**Instance-dependent episodic PAC RL.** The history of instance-dependent episodic PAC RL is not very long. Wagenmaker et al. (2022) proposed a planning-based algorithm, MOCA, and analyzed its instance-dependent sample complexity. Tirinzoni et al. (2022) provided an instance-dependent lower bound for deterministic MDPs and proposed the EPRL algorithm, which has an upper bound of sample complexity matching the lower bound up to a $H^2$ factor and logarithmic terms. Wagenmaker and Jamieson (2022) considered finite horizon linear MDPs, a superset of tabular MDPs. They proposed the PEDEL algorithm, which takes a policy set as an input, and analyzed its sample complexity. Tirinzoni et al. (2023) proved, for the first time, an instance-dependent sample complexity of an optimistic algorithm, BPI-UCRL.

**Instance-dependent pure exploration in multi-armed bandits.** The problem of pure exploration in multi-armed bandits (a special case of RL with $S = H = 1$) has a rich history and is typically studied in two frameworks: the fixed-confidence (($\varepsilon, \delta$)-PAC) setting and the fixed-budget setting.

In the fixed-confidence setting, the goal is to identify an arm whose mean reward is within $\varepsilon$ of the optimal arm's mean with probability at least $1 - \delta$, while minimizing the number of samples (pulls). Even-dar et al. (2002) initiated this line of work by proposing the Successive Elimination algorithm, which guarantees an optimal arm with probability $1 - \delta$ using distribution-dependent samples (($\varepsilon, \delta$)-sample complexity). Mannor and Tsitsiklis (2004) later provided a distribution-dependent lower bound on the ($\varepsilon, \delta$)-sample complexity. Kalyanakrishnan et al. (2012) proposed the LUCB algorithm and analyzed their sample complexity. Karnin et al. (2013) introduced the Exponential-Gap Elimination algorithm, removed unnecessary log factors and attained near-optimal sample complexity in the fixed-confidence regime. Garivier and Kaufmann (2016) gave a tighter lower bound and proposed an algorithm, Track and Stop, which exactly hits the lower bound asymptotically.

In the fixed-budget setting, the learner is given a total sampling budget $T$ and aims to maximize the probability of identifying the best arm by time $T$. Here, the results are often characterized by the exponential rate at which the failure probability decays with $T$. Audibert et al. (2010) studied this setting and proposed the Successive Rejects algorithm, proving that its error probability decays at an optimal rate, up to logarithmic factors in the number of arms. Karnin et al. (2013) proposed the Sequential Halving algorithm, proving that its error probability has an improved rate, which is optimal up to doubly logarithmic factors in the number of arms. Zhao et al. (2023) provided a tighter analysis of the Sequential Halving algorithm and obtained an accelerated decay rate of $\varepsilon$-error probability.

## B  PROPERTIES OF MDP

Although the statements and proofs of the lemmas in this section are nearly identical to those in the appendix of Wagenmaker et al. (2022), we include them here for completeness.

**Lemma 7.** *Assume that some deterministic policy $\hat{\pi}$ satisfies $\Delta_h^{\hat{\pi}}(s, \hat{\pi}_h(s)) \leq \varepsilon_h(s)$ for any $h' \leq h \leq H$ and any $s \in \mathcal{S}$. Then, for any policy $\pi'$,*

$$\sum_s w_{h'}^{\pi'}(s) \left( V_{h'}^*(s) - V_{h'}^{\hat{\pi}}(s) \right) \leq \sum_{h=h'}^{H} \sup_{\pi} \sum_s w_h^{\pi}(s) \varepsilon_h(s).$$

*Proof.* The proof proceeds by backward induction on $h'$. When $h' = H$, the statement trivially holds. Assume that

$$\sum_s w_{h'}^{\pi'}(s) \left( V_{h'}^*(s) - V_{h'}^{\hat{\pi}}(s) \right) \leq \sum_{h=h'}^{H} \sup_{\pi} \sum_s w_h^{\pi}(s) \varepsilon_h(s)$$

holds for step $h' > 1$ and any policy $\pi$. Assume further that

$$\Delta_{h'-1}^{\hat{\pi}}(s, \hat{\pi}(s)) \leq \varepsilon_{h'-1}(s).$$

By definition,

$$
\begin{aligned}
V_{h'-1}^*(s) - V_{h'-1}^{\hat{\pi}}(s) &= Q_{h'-1}^*(s, \pi_{h'-1}^*(s)) - Q_{h'-1}^{\hat{\pi}}(s, \hat{\pi}_{h'-1}(s)) \\
&= \underbrace{Q_{h'-1}^*(s, \pi_{h'-1}^*(s)) - Q_{h'-1}^{\hat{\pi}}(s, \pi_{h'-1}^*(s))}_{(1)} + \underbrace{Q_{h'-1}^{\hat{\pi}}(s, \pi_{h'-1}^*(s)) - \max_a Q_{h'-1}^{\hat{\pi}}(s, a)}_{(2)} \\
&\quad + \underbrace{\max_a Q_{h'-1}^{\hat{\pi}}(s, a) - Q_{h'-1}^{\hat{\pi}}(s, \hat{\pi}_{h'-1}(s))}_{(3)}.
\end{aligned}
$$

It is obvious that $(2) \leq 0$ and $(3) = \Delta_{h'-1}^{\hat{\pi}}(s, \hat{\pi}_{h'-1}(s)) \leq \varepsilon_{h'-1}(s)$ by our assumption. Furthermore,

$$(1) = \sum_{s'} P_{h'-1}(s'|s, \pi_{h'-1}^*(s))(V_{h'}^*(s') - V_{h'}^{\hat{\pi}}(s')).$$

Then, for any policy $\pi'$,

$$
\begin{aligned}
\sum_s w_{h'-1}^{\pi'}(s)(V_{h'-1}^*(s) - V_{h'-1}^{\hat{\pi}}(s)) &\leq \sum_s \sum_{s'} w_{h'-1}^{\pi'}(s) P_{h'-1}(s'|s, \pi_{h'-1}^*(s))(V_{h'}^*(s') - V_{h'}^{\hat{\pi}}(s')) \\
&\quad + \sum_s w_{h'-1}^{\pi'}(s) \varepsilon_{h'-1}(s) \\
&= \sum_s w_{h'}^{\pi''}(s)(V_{h'}^*(s) - V_{h'}^{\hat{\pi}}(s)) + \sum_s w_{h'-1}^{\pi''}(s) \varepsilon_{h'-1}(s) \\
&\leq \sum_{h=h'-1}^{H} \sup_{\pi} \sum_s w_h^{\pi}(s) \varepsilon_h(s),
\end{aligned}
$$

where $\pi''$ is a policy that is equal to $\pi'$ in step $1, \dots, h'-2$ and equal to $\pi^*$ in step $h'-1, \dots, H$, the last inequality follows by the induction hypothesis. $\square$

**Lemma 8.** *Assume $\sup_{\pi} \sum_s w_{h+1}^{\pi}(s) \left( V_{h+1}^*(s) - V_{h+1}^{\hat{\pi}}(s) \right) \leq \varepsilon$. Then*

$$|\Delta_h(s, a) - \Delta_h^{\hat{\pi}}(s, a)| \leq \varepsilon / W_h(s).$$

*Proof.*

$$
\begin{aligned}
|\Delta_h(s, a) - \Delta_h^{\hat{\pi}}(s, a)| &= |V_h^*(s) - Q_h^*(s, a) - (\max_{a'} Q_h^{\hat{\pi}}(s, a') - Q_h^{\hat{\pi}}(s, a))| \\
&\leq \max\{|V_h^*(s) - \max_{a'} Q_h^{\hat{\pi}}(s, a')|, |Q_h^{\hat{\pi}}(s, a) - Q_h^*(s, a)|\},
\end{aligned}
$$

where the last inequality follows since

$$V_h^*(s) - Q_h^*(s,a) - (\max_{a'} Q_h^{\hat{\pi}}(s,a') - Q_h^{\hat{\pi}}(s,a)) \le V_h^*(s) - \max_{a'} Q_h^{\hat{\pi}}(s,a')$$

and

$$-(V_h^*(s) - Q_h^*(s,a) - (\max_{a'} Q_h^{\hat{\pi}}(s,a') - Q_h^{\hat{\pi}}(s,a))) \le Q_h^*(s,a) - Q_h^{\hat{\pi}}(s,a).$$

We can write

$$Q_h^*(s,a) = r_h(s,a) + \sum_{s'} P_h(s'|s,a) V_{h+1}^*(s'),$$

$$Q_h^{\hat{\pi}}(s,a) = r_h(s,a) + \sum_{s'} P_h(s'|s,a) V_{h+1}^{\hat{\pi}}(s').$$

Then we have

$$Q_h^*(s,a) - Q_h^{\hat{\pi}}(s,a) = \sum_{s'} P_h(s'|s,a)(V_{h+1}^*(s') - V_{h+1}^{\hat{\pi}}(s'))$$

$$= \frac{1}{W_h(s)} \sum_{s'} W_h(s) P_h(s'|s,a)(V_{h+1}^*(s') - V_{h+1}^{\hat{\pi}}(s'))$$

$$\le \frac{1}{W_h(s)} \sup_{\pi} \sum_{s'} w_{h+1}^{\pi}(s')(V_{h+1}^*(s') - V_{h+1}^{\hat{\pi}}(s')) \le \frac{\varepsilon}{W_h(s)}. \quad (2)$$

Let $a_1 := \arg\max_a Q_h^*(s,a)$. Then

$$V_h^*(s) - \max_{a'} Q_h^{\hat{\pi}}(s,a') = \max_{a'} Q_h^*(s,a') - \max_{a'} Q_h^{\hat{\pi}}(s,a') = Q_h^*(s,a_1) - \max_{a'} Q_h^{\hat{\pi}}(s,a')$$

$$= Q_h^*(s,a_1) - Q_h^{\hat{\pi}}(s,a_1) + Q_h^{\hat{\pi}}(s,a_1) - \max_{a'} Q_h^{\hat{\pi}}(s,a') \le \frac{\varepsilon}{W_h(s)}. \quad (3)$$

By (2), (3), the lemma follows. $\qquad\square$

## C  ANALYSIS OF FB-L2E

### C.1  ANALYSIS OF FINDEXPLORABLESETS

The overall analysis is similar to that of Wagenmaker et al. (2022). However, the details should be changed as we use STRONGEULER instead of EULER. We begin with a regret bound of STRONGEULER. Throughout this section, let $M := (SAH^2)^2$.

**Lemma 9.** *If we run* STRONGEULER *with confidence parameter $\delta$ for $K$ episodes, with probability at least $1 - \delta$,*

$$\sum_{k=1}^{K} V_0^* - \sum_{k=1}^{K} V_0^{\pi_k} \le c_{\mathrm{se}} \sqrt{SAH^2 V_0^* K \log(HK) \log(\frac{MHK}{\delta})} + c_{\mathrm{se}} S^2 AH^6 \log(HK) \log(\frac{MHK}{\delta}),$$

*where $M = (SAH^2)^2$ and $c_{\mathrm{se}}$ is a universal constant.*

*Proof.* In Simchowitz and Jamieson (2019, Theorem 2.4), the regret bound up to a universal constant is presented as

$$\sqrt{SA\bar{H}_T T \log(\frac{mT}{\delta})} + SAH^4(S \vee H) \log(\frac{mT}{\delta}) \min\{\log(\frac{mT}{\delta}), \log(\frac{mH}{\Delta_{\min}})\},$$

where $\Delta_{\min} = \min_{s,a,h}^+ \Delta_h(s,a)$, $T = HK$, $m = (SAH)^2$, and $\bar{H}_T \le \frac{\mathcal{G}^2}{H} \log(T)$. Here, $\mathcal{G}$ is a constant such that the reward of one episode of our MDP is bounded by $\mathcal{G}$. We can reduce this $\frac{\mathcal{G}^2}{H}$ term to $\frac{V_0^*}{4H}$ by using the argument used in the proof of Jin et al. (2020, Lemma 3.4) and Wagenmaker et al. (2022, Lemma D.4). Thus, the regret bound (up to a universal constant) of STRONGEULER is given as

$$\sqrt{SAV_0^* T \log(T) \log(\frac{mT}{\delta})} + SAH^4(S \vee H) \log(\frac{mT}{\delta}) \min\{\log(\frac{mT}{\delta}), \log(\frac{mH}{\Delta_{\min}})\},$$

The second term is derived from their Simchowitz and Jamieson (2019, Claim C.3). In the proof of Simchowitz and Jamieson (2019, Claim C.3), we can just bound

$$\log(1 + \frac{N \wedge n_{\text{end}}}{n_0}) \leq \log(1 + T)$$

since $N \leq T, n_0 \geq 1$. By using this bound, we get a regret bound of

$$\sqrt{SAV_0^* T \log(T) \log(\frac{mT}{\delta})} + SAH^4(S \vee H) \log(\frac{mT}{\delta}) \log(T).$$

Although this bound only applies to stationary MDPs, stationary MDPs can represent non-stationary MDPs by augmenting states $s$ to $(s, h)$. In this case, the effective number of states is $SH$. Thus, by substituting $SH$ in to $S$, $HK$ into $T$, the lemma follows. $\qquad\square$

We now define the important quantities

$$C_K(\delta, \delta_{\text{samp}}, i) := \max \left\{ 432c_{\text{se}}^2 S^3 A^2 H^6 (i+6)^2 \log^2(2 \cdot 2 \cdot 432c_{\text{se}}^2 S^3 A^2 H^7 M(i+6), \right.$$

$$432c_{\text{se}}^2 S^3 A^2 H^6 \log(\frac{1}{\delta})(i+3) \log(2 \cdot 432c_{\text{se}}^2 S^3 A^2 H^7 \log(\frac{1}{\delta})(i+3)), \tag{4}$$

$$\left. 24 \log(\frac{4}{\delta}), \quad 2^{11} S^2 A^2 \log(\frac{4SAH}{\delta_{\text{samp}}}) \right\},$$

$$K_i(\delta, \delta_{\text{samp}}) := \lceil 2^i C_K(\delta, \delta_{\text{samp}}, i) \rceil.$$

and prove the following property.

**Lemma 10.** *Let* $C_{\mathcal{R}} := 2c_{\text{se}} S^3 A^2 H^6 \log(HK_i) \log(\frac{2MHK_i}{\delta}) + 2\log\frac{4}{\delta}$ *and* $K_i = K_i(\delta, \delta_{\text{samp}})$. *Then,*

$$K_i \geq 2^i \max\{4C_{\mathcal{R}}, 144c_{\text{se}}^2 S^2 A^2 H^2 \log(HK_i) \log(\frac{2MHK_i}{\delta})\}.$$

*Proof.* For any $i, j > 0$ and $C > 0$, if $x \geq C^i(i+3j)^j \log^j(C(i+3j))$, then $x \geq C^i \log^j x$ since

$$\begin{aligned}
C^i \log^j x = C^i \log^j[C^i(i+3j)^j \log^j(C(i+3j))] &\leq C^i \log^j[C^{i+j}(i+3j)^{2j}] \\
&\leq C^i(i+3j)^j \log^j[C(i+3j)] \\
&= x
\end{aligned}$$

Since

$$2MHK_i \geq 2^i \cdot 2 \cdot 432c_{\text{se}}^2 S^3 A^2 H^7 M(i+6)^2 \log^2(2 \cdot 2 \cdot 432c_{\text{se}}^2 S^3 A^2 H^7 M(i+6),$$

we have

$$K_i \geq 2^i \cdot 2 \cdot 432c_{\text{se}}^2 S^3 A^2 H^6 \log^2(2MHK_i).$$

Since

$$HK_i \geq 2^i \cdot 432c_{\text{se}}^2 S^3 A^2 H^7 \log(\frac{1}{\delta})(i+3) \log(2 \cdot 432c_{\text{se}}^2 S^3 A^2 H^7 \log(\frac{1}{\delta})(i+3)),$$

we have

$$K_i \geq 2^i \cdot 432c_{\text{se}}^2 S^3 A^2 H^6 \log(HK_i) \log(\frac{1}{\delta}).$$

We also have $K_i \geq 2^i \cdot 24 \log(\frac{4}{\delta})$. Combining these three, we have

$$K_i \geq 2^i \left( 144c_{\text{se}}^2 S^3 A^2 H^6 (\log^2(2MHK_i) + \log(HK_i) \log(\frac{1}{\delta}) + 8\log(\frac{4}{\delta}) \right),$$

which easily implies

$$K_i \geq 2^i \cdot 144c_{\text{se}}^2 S^2 A^2 H^2 \log(HK_i) \log(\frac{2MHK_i}{\delta}),$$

$$K_i \geq 8^i \left( 2c_{\text{se}} S^3 A^2 H^6 \log(HK_i) \log(\frac{2MHK_i}{\delta}) + 8\log(\frac{4}{\delta}) \right) = 4C_{\mathcal{R}}.$$

$\square$

Throughout the rest of this subsection, we consider running

$$\text{FINDEXPLORABLESETS}(\mathcal{X}, h, \delta, K_i := K_i(\delta, \delta_{\text{samp}}), N_i := \frac{K_i}{4|\mathcal{X}|2^i})$$

(defined in Algorithm 1) with some $\mathcal{X} \subset \mathcal{S} \times \mathcal{A}$ satisfying

$$W_h(\mathcal{X}) \leq 2^{-i+1}.$$

Let $\mathcal{X}_i \subset \mathcal{X}$, $\Pi_i$ be the output. We introduce the following notations. Let $K_{ij}$ denote the total number of episodes taken for $j$, where the index $j$ changes when the reward $r_h^j$ is reset. Let $m_i$ denote the number of $j$. Thus, we have

$$\sum_{j=1}^{m_i} K_{ij} = K_i.$$

Let $V_0^{*,ij}$ denote the optimal value function on the reward function $r_h^j$, $V_0^{k,ij}$ denote the value function for the policy $\pi_k$ on the reward function $r_h^j$. Then,

$$V_0^{k,ij} \leq V_0^{*,ij} \leq \sup_\pi \mathbb{E}_\pi[\mathbb{I}\{(s_h, a_h) \in \mathcal{X}\}] = W_h(\mathcal{X}) \leq 2^{-(i-1)}.$$

Now we define some events.

$$\mathcal{C}_{1,\delta} = \Big\{ \sum_{j=1}^{m_i} \Big( \sum_{k=1}^{K_{ij}} V_0^{*,ij} - \sum_{k=1}^{K_{ij}} V_0^{k,ij} \Big) \leq 2c_{\text{se}}\sqrt{S^2 A^2 H^2 V_0^{*,i1} K_i \log(HK_i) \log(\frac{MHK_i}{\delta})}$$

$$+ 2c_{\text{se}}S^3 A^2 H^6 \log(HK_i) \log(\frac{MHK_i}{\delta}) \Big\},$$

$$\mathcal{C}_{2,\delta} = \Big\{ \Big| \sum_{j=1}^{m_i} \sum_{k=1}^{K_{ij}} \sum_{h=1}^{H} R_h^j(s_h^{j,k}, a_h^{j,k}) - \sum_{j=1}^{m_i} \sum_{k=1}^{K_{ij}} V_0^{k,ij} \Big| \leq \sqrt{4K_i 2^{-i} \log\frac{2}{\delta}} + 2\log\frac{2}{\delta} \Big\},$$

$$\mathcal{D}_{1,\delta} = \Big\{ \forall (s,a) \in \mathcal{X}, \Big| \sum_{k=1}^{K_i} w_h^{\pi_k}(s,a) - \sum_{k=1}^{K_i} \mathbb{I}_{\{(s_h^k, a_h^k) = (s,a)\}} \Big| \leq \sqrt{2K_i W_h(s) \log\frac{2}{\delta}} + 2\log\frac{2}{\delta} \Big\}$$

for the process during the algorithm,

$$\mathcal{D}_{2,\delta} = \Big\{ \forall (s,a) \in \mathcal{X}_i, \Big| \sum_{k=1}^{K_i} w_h^{\pi_k}(s,a) - \sum_{k=1}^{K_i} \mathbb{I}_{\{(s_h^k, a_h^k) = (s,a)\}} \Big| \leq \sqrt{2K_i W_h(s) \log\frac{2}{\delta}} + 2\log\frac{2}{\delta} \Big\}$$

for the process during the replay.

Freedman's inequality is stated below for use in subsequent analysis.

**Lemma 11** (Freedman's inequality). *Let $(\Omega, \mathcal{F}, \mathbb{P})$ be a probability space and $\mathcal{F}_0 \subset \mathcal{F}_1 \subset \mathcal{F}_2 \subset \cdots \mathcal{F}$ be a filtration of $\sigma$-algebra. Let $\{X_i\}_i$ be random variables such that $X_i$ is $\mathcal{F}_i$-measurable,*

$$|X_i| \leq M,$$
$$\mathbb{E}[X_n|\mathcal{F}_{n-1}] = 0,$$
$$\mathbb{E}[X_n^2|\mathcal{F}_{n-1}] \leq V_n$$

*for constants $V_n$. Then, for any $\delta > 0$, with probability at least $1 - \delta$,*

$$|\sum_{i=1}^{n} X_i| < 2M\log\frac{2}{\delta} + \sqrt{2\sum_{i=1}^{n} V_n \log\frac{2}{\delta}}.$$

We state properties of the events defined above.

**Lemma 12.** *If $\delta \in (0,1)$ is the third argument of* `FindExplorableSets`,

$$\mathbb{P}(\mathcal{C}_{1,\delta/2}) \geq 1 - \delta/2.$$

*Proof.* For any fixed $K$ and $j$,

$$\Big(\sum_{k=1}^{K} V_0^{*,ij} - \sum_{k=1}^{K} V_0^{k,ij}\Big)|\mathcal{F}_{j-1} \leq c_{\mathrm{se}}\sqrt{SAH^2 V_0^{*,i1} K \log(HK) \log(\frac{MHK}{\delta})}$$
$$+ c_{\mathrm{se}} S^2 A H^6 \log(HK) \log(\frac{MHK}{\delta})$$

with probability at least $1 - \delta$, where $\mathcal{F}_{j-1}$ is the filtration up to iteration $j$, and we used $V_0^{*,ij} \leq V_0^{*,i1}$ for all $j$ since the reward function can only decrease as $j$ increases. `FindExplorableSets` stops and restarts STRONGEULER if the relevant condition is met, but this is a random stopping condition. Thus, to guarantee that the regret bound holds for any possible value of this stopping time, we union bound over all possible values. Since `FindExplorableSets` runs for at most $K_i$ episodes, we union bound over $K_i$ stopping times. We then have

$$\Big(\sum_{k=1}^{K} V_0^{*,ij} - \sum_{k=1}^{K} V_0^{k,ij}\Big)|\mathcal{F}_{j-1} \leq 2c_{\mathrm{se}}\sqrt{SAH^2 V_0^{*,i1} K \log(HK_i) \log(\frac{2MHK_i}{\delta})}$$
$$+ 2c_{\mathrm{se}} S^2 A H^6 \log(HK_i) \log(\frac{2MHK_i}{\delta})$$

for all $K \in [K_i]$ with probability at least $1 - \frac{\delta}{2SA}$. Since $m_i \leq SA$, union bounding over all $j$ we then have that, with probability at least $1 - \delta/2$,

$$\sum_{j=1}^{m_i}\Big(\sum_{k=1}^{K_{ij}} V_0^{\star,ij} - \sum_{k=1}^{K_{ij}} V_0^{k,ij}\Big) \leq \sum_{j=1}^{m_i} 2c_{\mathrm{se}}\sqrt{SAH^2 V_0^{*,i1} K_{ij} \log(HK_i) \log(\frac{2MHK_i}{\delta})}$$
$$+ 2c_{\mathrm{se}} S^3 A^2 H^6 \log(HK_i) \log(\frac{2MHK_i}{\delta})$$
$$\leq 2c_{\mathrm{se}}\sqrt{S^2 A^2 H^2 V_0^{*,i1} K_i \log(HK_i) \log(\frac{2MHK_i}{\delta})}$$
$$+ 2c_{\mathrm{se}} S^3 A^2 H^6 \log(HK_i) \log(\frac{2MHK_i}{\delta}),$$

where the last inequality follows from Jensen's inequality. $\square$

**Lemma 13.** *For any $\delta \in (0,1)$,*

$$\mathbb{P}(\mathcal{C}_{2,\delta}) \geq 1 - \delta.$$

*Proof.* For each $k \in [K_i]$, we have that $X_k := \sum_{h=1}^{H} R_h(s_h^k, a_h^k) \sim \mathrm{Bernoulli}(V_0^{\pi_k})$. Then $|X_k - V_0^{\pi_k}| \leq 1$, $\mathbb{E}[(X_k - V_0^{\pi_k})^2|\mathcal{F}_{k-1}] = V_0^{\pi_k}(1 - V_0^{\pi_k}) \leq V_0^{\pi_k} \leq W_h(\mathcal{X}) \leq 2^{-i+1}$. Thus, if we apply Lemma 11, we obtain the statement. $\square$

**Lemma 14.** *For any $\delta \in (0,1)$,*

$$\mathbb{P}(\mathcal{D}_{1,\delta}) \geq 1 - |\mathcal{X}|\delta \geq 1 - SA\delta.$$

*Proof.* Since $X_k := \mathbb{I}_{\{(s_h^k, a_h^k)=(s,a)\}} \sim \mathrm{Bernoulli}(w_h^{\pi_k}(s,a))$,

$$\mathbb{E}[(X_k - w_h^{\pi_k}(s,a))^2|\mathcal{F}_{k-1}] = w_h^{\pi_k}(s,a)(1 - w_h^{\pi_k}(s,a)) \leq w_h^{\pi_k}(s,a) \leq W_h(s).$$

By Lemma 11, we have that

$$\left|\sum_{k=1}^{K_i} w_h^{\pi_k}(s,a) - \sum_{k=1}^{K_i} \mathbb{I}_{\{(s_h^k, a_h^k)=(s,a)\}}\right| \leq \sqrt{2K_i W_h(s) \log\frac{2}{\delta}} + 2\log\frac{2}{\delta}$$

with probability at least $1 - \delta$. Union bounding over $\mathcal{X}$ leads to the statement. $\square$

**Lemma 15.** *For any $\delta \in (0, 1)$,*

$$\mathbb{P}(\mathcal{D}_{2,\delta}) \geq 1 - |\mathcal{X}_i|\delta \geq 1 - SA\delta.$$

*Proof.* Since $X_k := \mathbb{I}_{\{(s_h^k, a_h^k) = (s,a)\}} \sim \text{Bernoulli}(w_h^{\pi_k}(s, a))$,

$$\mathbb{E}[(X_k - w_h^{\pi_k}(s, a))^2 | \mathcal{F}_{k-1}] = w_h^{\pi_k}(s, a)(1 - w_h^{\pi_k}(s, a)) \leq w_h^{\pi_k}(s, a) \leq W_h(s).$$

By Lemma 11 and union bound over $\mathcal{X}_i$, the statement follows. $\qquad\square$

**Lemma 16.** *If $\delta \in (0, 1)$ is the third argument of* `FindExplorableSets`*, the event $\mathcal{C}_{1,\delta/2} \cap \mathcal{C}_{2,\delta/2}$ implies*

$$W_h(\mathcal{X} \setminus \mathcal{X}_i) \leq 2^{-i}.$$

*Proof.* Putting Lemma 12, 13 and union bounding over these events, we have that with probability at least $1 - \delta$,

$$\sum_{j=1}^{m_i}\sum_{k=1}^{K_{ij}}\sum_{h=1}^{H} R_h^j(s_h^{j,k}, a_h^{j,k}) \geq \sum_{j=1}^{m_i}\sum_{k=1}^{K_{ij}} V_0^{\star,ij} - \sqrt{4K_i 2^{-i} \log\frac{4}{\delta}}$$

$$- 2c_{\text{se}}\sqrt{S^2 A^2 H^2 V_0^{*,i1} K_i \log(HK_i) \log(\frac{2MHK_i}{\delta})} - C_{\mathcal{R}}$$

where we denote

$$C_{\mathcal{R}} := 2c_{\text{se}}S^3 A^2 H^6 \log(HK_i) \log(\frac{2MHK_i}{\delta}) + 2\log\frac{4}{\delta}.$$

Assume that $V_0^{*,im_i} > 2^{-i}$. Using that the reward decreases monotonically so $V_0^{*,im_i} \leq V_0^{*,ij}$ for any $j \leq m_i$, we can lower bound the above as

$$\geq 2^{-i}K_i - \sqrt{4K_i 2^{-i} \log\frac{4}{\delta}} - 2c_{\text{se}}\sqrt{S^2 A^2 H^2 V_0^{*,i1} K_i \log(HK_i) \log(\frac{2MHK_i}{\delta})} - C_{\mathcal{R}}$$

$$\geq 2^{-i}K_i - 3c_{\text{se}}\sqrt{S^2 A^2 H^2 2^{-i} K_i \log(HK_i) \log(\frac{2MHK_i}{\delta})} - C_{\mathcal{R}}$$

where the second inequality follows since $V_0^{*,i1} \leq 2^{-i+1}$ and $\sqrt{4K_i 2^{-i} \log\frac{4}{\delta}}$ will then be dominated by the regret term. Lemma 10 gives

$$K_i \geq 2^i \max\left\{4C_{\mathcal{R}}, 144c_{\text{se}}^2 S^2 A^2 H^2 \log(HK_i) \log(\frac{2MHK_i}{\delta})\right\}$$

which implies

$$\frac{1}{4}2^{-i}K_i - C_{\mathcal{R}} \geq 0$$

and

$$\frac{1}{4}2^{-i}K_i - 3c_{\text{se}}\sqrt{S^2 A^2 H^2 2^{-i} K_i \log(HK_i) \log(\frac{2MHK_i}{\delta})}$$

$$\geq \frac{2^i \cdot 144c_{\text{se}}^2 S^2 A^2 H^2 \log(HK_i) \log(\frac{2MHK_i}{\delta})}{4 \cdot 2^i}$$

$$- 3c_{\text{se}}\sqrt{S^2 A^2 H^2 2^{-i} \log(HK_i) \log(\frac{2MHK_i}{\delta}) \cdot 2^i 144c_{\text{se}}^2 S^2 A^2 H^2 \log(HK_i) \log(\frac{2MHK_i}{\delta})}$$

$$= 0.$$

Thus, we can lower bound the above as

$$2^{-i}K_i - 3c_{\text{se}}\sqrt{S^2 A^2 H^2 2^{-i} K_i \log(HK_i) \log(\frac{2MHK_i}{\delta})} - C_{\mathcal{R}} \geq \frac{1}{2}2^{-i}K_i.$$

Note that we can collect a total reward of at most $|\mathcal{X}|N_i$. However, by our choice of

$$N_i = K_i/(4|\mathcal{X}| \cdot 2^i),$$

we have that

$$|\mathcal{X}|N_i = \frac{1}{4 \cdot 2^i} K_i < \frac{1}{2 \cdot 2^i} K_i.$$

This is a contradiction. Thus, we must have that $W_h(\mathcal{X} \setminus \mathcal{X}_i) \le V_0^{*,im_i} \le 2^{-i}$. $\qquad\square$

**Lemma 17.** *The event $\mathcal{C}_{\delta/2}$ with $\delta \ge \frac{\delta_{\text{samp}}}{SAH}$ implies*

$$W_h(\mathcal{X}) \ge \frac{|\mathcal{X}_i|}{2^{i+3}|\mathcal{X}|}.$$

*Proof.*

$$N_i|\mathcal{X}_i| \le \sum_{j=1}^{m_i}\sum_{k=1}^{K_{ij}} R_h^j(s_h^{j,k}, a_h^{j,k}) \le \sum_{j=1}^{m_i}\sum_{k=1}^{K_{ij}} V_0^{k,ij} + \sqrt{4K_i 2^{-i}\log\frac{4}{\delta}} + 2\log\frac{4}{\delta}$$

$$\le K_i W_h(\mathcal{X}) + \sqrt{4K_i 2^{-i}\log\frac{4}{\delta}} + 2\log\frac{4}{\delta}$$

$$\le K_i W_h(\mathcal{X}) + \frac{K_i}{2^{i+4}SA} + \frac{K_i}{2^{i+10}SA}$$

$$\le K_i W_h(\mathcal{X}) + \frac{K_i}{2^{i+3}SA},$$

where the forth inequality follows from $K_i \ge 2^{i+11}S^2A^2\log\frac{4SAH}{\delta_{\text{samp}}}$. Then,

$$W_h(\mathcal{X}) \ge \frac{N_i|\mathcal{X}_i|}{K_i} - \frac{1}{2^{i+3}SA} = \frac{|\mathcal{X}_i|}{2^{i+2}|\mathcal{X}|} - \frac{1}{2^{i+3}SA} \ge \frac{|\mathcal{X}_i|}{2^{i+3}|\mathcal{X}|}.$$

$\qquad\square$

**Lemma 18.** *The event $\mathcal{D}_{1,\delta} \cap \mathcal{D}_{2,\delta}$ with $\delta \ge \frac{\delta_{\text{samp}}}{2SAH}$ implies that after rerunning each policy in $\Pi_i$ once, the number of samples collected for each $(s,a) \in \mathcal{X}_i$ is at least $\frac{1}{4}N_i$.*

*Proof.* Let $\mathbb{I}^1, \mathbb{I}^2$ denote the indicator of an event during `FindExplorableSets`, and an event during rerunning policies respectively. For a pair $(s,a) \in \mathcal{X}_i$, we have

$$\sum_{k=1}^{K_i} \mathbb{I}^1_{\{(s_h^k, a_h^k)=(s,a)\}} - \sum_{k=1}^{K_i} w_h^{\pi_k}(s,a) \le \sqrt{2K_i W_h(s)\log\frac{2}{\delta}} + 2\log\frac{2}{\delta}$$

$$\sum_{k=1}^{K_i} w_h^{\pi_k}(s,a) - \sum_{k=1}^{K_i} \mathbb{I}^2_{\{(s_h^k, a_h^k)=(s,a)\}} \le \sqrt{2K_i W_h(s)\log\frac{2}{\delta}} + 2\log\frac{2}{\delta}$$

Then the number of samples of $(s,a)$ collected during the rerunning satisfies

$$\sum_{k=1}^{K_i} \mathbb{I}^2_{\{(s_h^k, a_h^k)=(s,a)\}} \ge \sum_{k=1}^{K_i} \mathbb{I}^1_{\{(s_h^k, a_h^k)=(s,a)\}} - 2\sqrt{2K_i W_h(s)\log\frac{2}{\delta}} - 4\log\frac{2}{\delta}$$

$$\ge N_i - 2\sqrt{2K_i W_h(s)\log\frac{2}{\delta}} - 4\log\frac{2}{\delta}$$

$$\ge N_i - 2\sqrt{2^{-i+2}K_i\log\frac{2}{\delta}} - 4\log\frac{2}{\delta}$$

$$\ge N_i - \frac{K_i}{2^{i+3.5}SA} - \frac{K_i}{2^{i+9}S^2A^2}$$

$$\ge N_i - \frac{K_i}{2^{i+2.5}SA}$$

$$\geq N_i - \frac{K_i}{2^{i+2.5}|\mathcal{X}|} = N_i(1 - \frac{1}{\sqrt{2}}) \geq \frac{1}{4}N_i,$$

where the forth inequality follows from $\delta \geq \frac{\delta_{\text{samp}}}{4SAH}$ and $\log \frac{2SAH}{\delta_{\text{samp}}} \leq \frac{K_i}{2^{i+11}S^2A^2}$. $\qquad\square$

**Lemma 19.** *The event $\mathcal{D}_{1,\delta}$ with $\delta \geq \frac{\delta_{\text{samp}}}{2SAH}$ implies*

$$W_h(s) > \frac{1}{2^{i+3}|\mathcal{X}|} \text{ for each } (s,a) \in \mathcal{X}_i, \quad W_h(\mathcal{X}_i) > \frac{|\mathcal{X}_i|}{2^{i+3}|\mathcal{X}|}.$$

*Proof.* In the proof of the previous lemma, we showed that

$$\sqrt{2^{-i+2}K_i \log \frac{2}{\delta}} + 2\log \frac{2}{\delta} \leq \frac{N_i}{2\sqrt{2}} < \frac{N_i}{2}$$

when $\delta \geq \frac{\delta_{\text{samp}}}{2SAH}$. Using this, we have

$$N_i \leq \sum_{k=1}^{K_i} \mathbb{I}^1_{\{(s_h^k, a_h^k)=(s,a)\}} \leq \sum_{k=1}^{K_i} w_h^{\pi_k}(s,a) + \sqrt{2^{-i+2}K_i \log \frac{2}{\delta}} + 2\log \frac{2}{\delta} < K_i W_h(s) + \frac{N_i}{2}$$

for each $(s,a) \in \mathcal{X}_i$. Thus,

$$W_h(s) > \frac{N_i}{2K_i} = \frac{1}{2^{i+3}|\mathcal{X}|}.$$

On the other hand,

$$|\mathcal{X}_i|N_i \leq \sum_{(s,a)\in\mathcal{X}_i} \sum_{k=1}^{K_i} \mathbb{I}^1_{\{(s_h^k, a_h^k)=(s,a)\}} \leq \sum_{k=1}^{K_i} w_h^{\pi_k}(\mathcal{X}_i) + |\mathcal{X}_i| \left( \sqrt{2^{-i+2}K_i \log \frac{2}{\delta}} + 2\log \frac{2}{\delta} \right) < K_i W_h(\mathcal{X}_i) + \frac{|\mathcal{X}_i|N_i}{2}.$$

Thus,

$$W_h(\mathcal{X}_i) > \frac{|\mathcal{X}_i|N_i}{2K_i} = \frac{|\mathcal{X}_i|}{2^{i+3}|\mathcal{X}|}.$$

$\qquad\square$

We finally give a guarantee of `FindExplorableSets`.

**Theorem C.1.** *If we run*

$$\texttt{FindExplorableSets}(\mathcal{X}, h, \delta, K_i = K_i(\delta, \delta_{\text{samp}} = SAH\delta), N_i = \frac{K_i}{4|\mathcal{X}|2^i})$$

*for a subset $\mathcal{X} \subset \mathcal{S} \times \mathcal{A}$ with $W_h(\mathcal{X}) \leq 2^{-i+1}$ and returns subset $\mathcal{X}_i \subset \mathcal{X}$, policy set $\Pi_i$, then*

1. *$W_h(\mathcal{X} \setminus \mathcal{X}_i) \leq 2^{-i}$ with probability at least $1 - \delta$.*

2. *With probability at least $1 - SA\delta$,*

    (1) *If we rerun each policy in $\Pi_i$ once, the number of samples collected for each $(s,a) \in \mathcal{X}_i$ is at least $\frac{1}{4}N_i$.*

    (2) *$W_h(s) > \frac{1}{2^{i+3}|\mathcal{X}|}$ for each $(s,a) \in \mathcal{X}_i$ and $W_h(\mathcal{X}_i) > \frac{|\mathcal{X}_i|}{2^{i+3}|\mathcal{X}|}$.*

*Proof.* By Lemma 12, 13, 14, 15, 16, 18, and 19, the theorem follows. $\qquad\square$

## C.2 PROOF OF THEOREM 3.1

Before proving Theorem 3.1, we introduce a useful lemma related to the Lambert $W$-function. The Lambert function $W(s) : [0, \infty) \to [0, \infty)$ is defined by

$$x = W(x) \exp(W(x)), \quad \text{for } x \geq 0.$$

Then the following holds.

**Lemma 20.** *(Orabona and Pal, 2016, Lemma 17)*

$$0.6321 \log(1 + x) \leq W(x) \leq \log(1 + x) \text{ for } x \geq 0.$$

We define

$$c(B) = 4JC_K(\frac{1}{8SAH}, \frac{1}{8}, J) = \text{poly}(S, A, H, \log(B)),$$
$$C_{\text{L2E}}(B) = SH^2 c(B). \tag{5}$$

Recall that $C_K$ was defined in equation 4. We now give a proof of Theorem 3.1

**Theorem C.2** (Theorem 3.1). *Consider running Algorithm 1 with $B \geq c(B)$. Then, the following statements hold.*

1. *The total budget used is at most $B$.*

2. *For any $\varepsilon \geq 2SH^2 \varepsilon_B$, with probability at least $1 - \exp\left(-\tilde{\Theta}\left(\frac{\varepsilon B}{C_{\text{L2E}}(B)}\right)\right)$,*

    *(1) The reachability of each set $\mathcal{X}_i$ satisfies*

    $$\frac{|\mathcal{X}_i|}{|\mathcal{X}|} \cdot 2^{-i-3} \leq W_h(\mathcal{X}_i) \leq 2^{-i+1} \quad \text{for all } i \leq i_\varepsilon := \left\lceil \log_2\left(\frac{2SH^2}{\varepsilon}\right)\right\rceil,$$

    *(2) The remaining elements, $\bar{\mathcal{X}} := \mathcal{X} \setminus \cup_{i=1}^{i_\varepsilon} \mathcal{X}_i$ satisfy*

    $$\sup_\pi \sum_{(s,a) \in \bar{\mathcal{X}}} w_h^\pi(s, a) \leq \frac{\varepsilon}{2SH^2}.$$

    *(3) Moreover, for any $i \leq i_\varepsilon$, if each policy in $\Pi_i$ is executed $A$ times, then every state-action pair $(s, a) \in \mathcal{X}_i$ is visited at least $\frac{1}{8} A N_i$ times.*

    *Here, the probability accounts for both the randomness in the execution of the algorithm and the resampling process.*

*Proof.* We first prove that the total budget used is at most $B$. Let $\delta = \frac{1}{8SAH}$. By the definition of $\delta_i$,

$$\log \frac{1}{\delta_i} = 0.6321 L_i \log \frac{1}{\delta} \cdot \log \log \frac{1}{\delta}$$

$$\leq 1 + 0.6321 L_i \log \frac{1}{\delta} \cdot \log \log \frac{1}{\delta}$$

$$\leq (1 + L_i \log \frac{1}{\delta} \cdot \log \log \frac{1}{\delta})^{0.6321}$$

$$\leq \exp(W(L_i \log \frac{1}{\delta} \cdot \log \log \frac{1}{\delta})).$$

Thus,

$$\log \frac{1}{\delta_i} \cdot \log \log \frac{1}{\delta_i} \leq W(L_i \log \frac{1}{\delta} \cdot \log \log \frac{1}{\delta}) \exp(W(L_i \log \frac{1}{\delta} \cdot \log \log \frac{1}{\delta})) = L_i \log \frac{1}{\delta} \cdot \log \log \frac{1}{\delta}. \tag{6}$$

The total budget used is

$$\sum_{j=1}^{J} K_j(\delta_j, SAH\delta_j) \leq \sum_{j=1}^{J} 2^{j+1} C_K(\delta_j, SAH\delta_j, J)$$

$$\leq \sum_{j=1}^{J} 2^{J+1} C_K(\frac{1}{8SAH}, \frac{1}{8}, J)$$

$$\leq 2J(1 + \frac{\log(2)B}{c(B)})^{0.6321} C_K(\frac{1}{8SAH}, \frac{1}{8}, J)$$

$$\leq 2J(1 + \frac{B}{c(B)}) C_K(\frac{1}{8SAH}, \frac{1}{8}, J),$$

where the second inequality follows from equation 6 and that $C_K$ has $\log(\frac{1}{\delta}) \log \log(\frac{1}{\delta})$ dependence. If $B \geq c(B)$, then the above is bounded by

$$\frac{4JB}{c(B)} C_K(\frac{1}{8SAH}, \frac{1}{8}, J) = B.$$

We now prove the second part. By union bounding Theorem C.1 over $i = 1, 2, \ldots i_\varepsilon$, (1) hold with probability at least

$$1 - \sum_{i=1}^{i_\varepsilon} \delta_i \geq 1 - i_\varepsilon \delta_{i_\varepsilon}.$$

Here, $\delta_{i_\varepsilon} = \exp(-\tilde{\Theta}(L_{i_\varepsilon}))$ by the definition and

$$L_{i_\varepsilon} = 2^{J-i_\varepsilon} \geq \frac{\varepsilon}{4SH^2}(1 + \frac{\log(2)B}{c(B)})^{0.6321} \geq \frac{\varepsilon}{4SH^2}(1 + 0.6321\frac{\log(2)B}{c(B)}) \geq \frac{\varepsilon}{4SH^2} \cdot 0.6321\frac{\log(2)B}{c(B)}.$$

Thus, (1) holds with probability at least $1 - \exp\left(-\tilde{\Theta}\left(\frac{\varepsilon B}{C_{\text{L2E}}(B)}\right)\right)$. Similarly, (2) holds with probability at least

$$1 - SA \sum_{i=1}^{i_\varepsilon} \delta_i.$$

Since $SA$ becomes $\log(SA)$ when moving into the exponential, (2) also holds with probability at least $1 - \exp\left(-\tilde{\Theta}\left(\frac{\varepsilon B}{C_{\text{L2E}}(B)}\right)\right)$. We next compute the probability that (3) holds. For simplicity, let's consider the level $i = i_\varepsilon$, in which the failure probability $SA\delta_{i_\varepsilon}$ is dominant. For the collection of samples via rerunning policies to be successful, we need both $\mathcal{D}_{1,\delta_{i_\varepsilon}}$ and $\mathcal{D}_{2,\delta_{i_\varepsilon}}$ to hold. $\mathcal{D}_{1,\delta_{i_\varepsilon}}$ holds with probability at least $1 - \frac{SA\delta_{i_\varepsilon}}{2}$. On the event $\mathcal{D}_{1,\delta_{i_\varepsilon}}$, consider rerunning each policy in $\Pi_{i_\varepsilon}$ for $A$ times. By Lemma 21, with probability $1 - \exp(-\frac{1}{2}A\log(\frac{1}{eSA\delta_{i_\varepsilon}}))$, at least for $\frac{A}{2}$ trials of repetition, we collect $\frac{N_{i_\varepsilon}}{4}$ samples of each $(s,a) \in \mathcal{X}_{i_\varepsilon}$, which means we collect at least $\frac{AN_{i_\varepsilon}}{8}$ samples of each $(s,a) \in \mathcal{X}_{i_\varepsilon}$. Thus, the probability that there exists some $(s,a) \in \mathcal{X}_{i_\varepsilon}$, the sample number of which is less than $\frac{AN_{i_\varepsilon}}{8}$ is

$$\exp\left(-\frac{1}{2}A\log(\frac{1}{eSA\delta_{i_\varepsilon}})\right) = \exp\left(-\tilde{\Theta}\left(\frac{\varepsilon AB}{C_{\text{L2E}}(B)}\right)\right).$$

However, the failure probability of $\mathcal{D}_{1,\delta_{i_\varepsilon}}$ is already $\exp\left(-\tilde{\Theta}\left(\frac{\varepsilon B}{C_{\text{L2E}}(B)}\right)\right)$, which is more dominant.

Thus, (3) also holds with probability $1 - \exp\left(-\tilde{\Theta}\left(\frac{\varepsilon B}{C_{\text{L2E}}(B)}\right)\right)$. The theorem is proven. $\qquad\square$

## C.3 BOOSTING TECHNIQUE

In this subsection, we develop an alternative algorithm of FB-L2E. The core mechanism of this alternative is the boosting technique, which repeatedly executes independent trials. The number of repetitions and the failure probability is in the exponential relationship as we can see in the following lemma.

**Lemma 21.** *Let $\mathcal{E}$ be an event from a random trial such that $\mathbb{P}(\mathcal{E}) \leq \delta$ Let $\alpha \in (\delta, 1)$. Let $N$ be the number of trials where $\mathcal{E}$ is true out of $L$ trials. Assume $\alpha > \delta$. Then,*

$$\mathbb{P}(\frac{N}{L} \geq \alpha) \leq \exp\left(-\alpha L \ln\left(\frac{\alpha}{e\delta}\right)\right)$$

*Proof.* Recall the KL divergence based concentration inequality where $\hat{\mu}_n$ is the sample mean of $n$ Bernoulli i.i.d. random variables with head probability $\mu$:

$$\mathbb{P}(\hat{\mu}_n - \mu \geq \varepsilon) \leq \exp(-n\mathsf{kl}(\mu + \varepsilon, \mu)) .$$

Note that $N/L$ can be viewed as the sample mean of Bernoulli trials with $\mu := \mathbb{P}(\mathcal{E})$. Then,

$$
\begin{aligned}
\mathbb{P}(N \geq \alpha L) = \mathbb{P}(\frac{N}{L} \geq \alpha) \\
= \mathbb{P}(\frac{N}{L} - \mu \geq \alpha - \mu) \\
\leq \exp(-L\mathsf{kl}(\alpha, \mu)) \\
= \exp\left(-L\left(\alpha \ln(\alpha/\mu) + (1-\alpha)\ln\frac{1-\alpha}{1-\mu}\right)\right) \\
\overset{(a)}{\leq} \exp\left(-L\left(\alpha \ln(\alpha/\mu) - \alpha\right)\right) \\
\leq \exp\left(-L\left(\alpha \ln(\alpha/\delta) - \alpha\right)\right)
\end{aligned}
$$

where $(a)$ is by the following derivation:

$$
\begin{aligned}
(1-\alpha)\ln\frac{1-\alpha}{1-\mu} = -(1-\alpha)\ln\frac{1-\mu}{1-\alpha} \\
= -(1-\alpha)\ln\left(1 + \frac{\alpha - \mu}{1-\alpha}\right) \\
\geq -(\alpha - \mu) \\
\geq -\alpha
\end{aligned}
$$

$\square$

The alternative algorithm, FB-L2E-BS is described in Algorithm 0. Although it only applies to singleton subsets (subset of size 1), one can flexibly change the regret minimization algorithm in FINDEXPLORABLESETS. It was crucial for our result that the regret bound of STRONGEULER has $\log(\frac{1}{\delta})$ dependence. However, for FB-L2E-BS, we can use algorithms such as EULER, which has $\log^3(\frac{1}{\delta})$ dependence in the lower order term.

We briefly argue that the statements of Theorem 3.1 also hold for FB-L2E-BS used for singleton subset. The total budget used is

$$
\begin{aligned}
\sum_{j=1}^{J} 2^{J-j}K_j = J2^{J+1}C_K(\frac{1}{8SAH}, \frac{1}{8}, J) \\
\leq 2J(1 + \frac{\log(2)B}{c(B)})^{0.6321}C_K(\frac{1}{8SAH}, \frac{1}{8}, J) \\
\leq 2J(1 + \frac{\log(2)B}{c(B)})C_K(\frac{1}{8SAH}, \frac{1}{8}, J).
\end{aligned}
$$

If $B \geq c(B)$, then the above is bounded by

$$\frac{4JB}{c(B)}C_K(\frac{1}{8SAH}, \frac{1}{8}, J) = B.$$

---

**Algorithm 4** **F**ixed **B**udget **L**earn to **E**xplore with **B**oosting for **S**ingleton (FB-L2E-BS)

> **function** FB-L2E-BS($\mathcal{X} = \{(s,a)\} \subseteq \mathcal{S} \times \mathcal{A}$, step $h$, budget $B$)
>> **if** $|\mathcal{X}| = 0$ **then**
>>> **return** $\{(\emptyset, \emptyset, 0,)\}$
>>
>> **end if**
>> $J \leftarrow \lceil 0.6321 \log_2(1 + \frac{\log(2)B}{c(B)}) \rceil$
>> **for** $j = 1, \ldots, J$ **do**
>>> $K_j \leftarrow K_j(\frac{1}{8SAH}, \frac{1}{8}), \quad N_j \leftarrow K_j/(4|\mathcal{X}| \cdot 2^j), \quad L_j \leftarrow 2^{J-j}$
>>> **for** $m = 1, \ldots, L_j$ **do**
>>>> $\mathcal{Y}_{j,m}, \Pi_{j,m} = \texttt{FindExplorableSets}(\mathcal{X}, h, \frac{1}{8SAH}, K_j, N_j)$
>>>
>>> **end for**
>>> Calculate the votes: $\forall (s,a) \in \mathcal{X}, v_{s,a} \leftarrow \sum_{m=1}^{L_j} \mathbb{1}\{(s,a) \in \mathcal{Y}_{j,m}\}$.
>>> Filter out only if chosen at least half the time: $\mathcal{X}_j \leftarrow \{(s,a) \mid v_{s,a} \geq L_j/2\}$
>>> $\Pi_j = \cup_{m=1}^{L_j} \Pi_{j,m}$
>>> $\mathcal{X} \leftarrow \mathcal{X} \backslash \mathcal{X}_j$
>>
>> **end for**
>> **return** $\{(\mathcal{X}_j, \Pi_j, N_j)\}_{j=1}^J$
>
> **end function**

---

Let $\delta = \frac{1}{8SAH}, \delta_{\text{samp}} = \frac{1}{8}$. The crucial part for other statements in Theorem 3.1, was to make the failure probability of the $j$-th iteration in the form of

$$(c_1 \delta)^{c_2 L_j} \tag{7}$$

for some constant $c_1, c_2$, which was done by defining $\delta_i$ as this form in FB-L2E. Once we get equation 7, the dominant term becomes $(c_1 \delta)^{c_2 L_{i_\varepsilon}} = \exp\left(-\tilde{\Theta}\left(\frac{\varepsilon B}{C_{\text{L2E}}(B)}\right)\right)$. We show that equation 7 can also be obtained for FB-L2E-BS.

Assume $W_h(s) \in (2^{-i}, 2^{-i+1}]$ for $i \leq J$. Let's call $i$ as the *reachable index* of $s$ at $h$. Let $\mathcal{N}_j$ be the event that $(s,a)$ is not filtered in $j$-th boosted FES. By Lemma 16,

$$\mathbb{P}\left((s,a) \text{ is not filtered in } i\text{-th step by a single FES} \mid \cap_{j=1}^{i-1} \mathcal{N}_j\right) \leq \delta.$$

If we apply Lemma 21, we obtain the form of equation 7 as

$$\mathbb{P}\left(\cap_{j=1}^i \mathcal{N}_j\right) \leq \mathbb{P}\left(\mathcal{N}_i \mid \cap_{j=1}^{i-1} \mathcal{N}_j\right) \leq \exp\left(-\frac{1}{2} L_i \log \frac{1}{2e\delta}\right).$$

We say that $(s,a)$ is *upper well-filtered* at $h$ if $(s,a)$ is filtered in the index $j$ for some $j \leq i$.

Now we consider the $j$-th boosted FES for some $j \leq i - 4$. By Lemma 14, 19,

$$\mathbb{P}\left((s,a) \text{ is filtered in } j\text{-th step by a single FES} \mid \cap_{k=1}^{j-1} \mathcal{N}_k\right) \leq \frac{\delta_{\text{samp}}}{2SAH}.$$

Thus, by Lemma 21, we obtain the form of equation 7 as

$$\mathbb{P}\left(\cap_{k=1}^{j-1} \mathcal{N}_k, \quad \mathcal{N}_j^{\mathsf{c}}\right) \leq \mathbb{P}\left(\mathcal{N}_j^{\mathsf{c}} \mid \cap_{k=1}^{j-1} \mathcal{N}_k\right) \leq \exp\left(-\frac{1}{2} L_j \log \frac{SAH}{e\delta_{\text{samp}}}\right).$$

We say that $(s,a)$ is *lower well-filtered* at $h$ if $(s,a)$ is not filtered in the indices $j$ with $j \leq i - 4$. We also say that $(s,a)$ is *well-filtered* at $h$ if $(s,a)$ is both upper and lower well-filtered at $h$. We have

$$\mathbb{P}\left((s,a) \text{ is not lower well-filtered at } h\right) \leq \sum_{j=1}^{i-4} \exp\left(-\frac{1}{2} L_j \log \frac{SAH}{e\delta_{\text{samp}}}\right) \leq i \exp\left(-\frac{1}{2} L_i \log \frac{SAH}{e\delta_{\text{samp}}}\right).$$

Thus, we have

$$\mathbb{P}\left((s,a) \text{ is well-filtered at } h\right) \geq 1 - \exp\left(-\frac{1}{2} L_i \log \frac{1}{2e\delta}\right) - i \exp\left(-\frac{1}{2} L_i \log \frac{SAH}{e\delta_{\text{samp}}}\right).$$

Recall that $\varepsilon \geq 2SH^2 \varepsilon_B$ and $i_\varepsilon := \lceil \log_2(\frac{2SH^2}{\varepsilon}) \rceil$. We define the set

$$\mathcal{S}_\varepsilon = \{(s, h) : \text{the reachable index of } s \text{ at } h \leq i_\varepsilon\}$$

and the event

$$\mathcal{M}_\varepsilon = \{(s, a) \text{ is well-filtered at } h \text{ for all } (s, h) \in \mathcal{S}_\varepsilon\}.$$

By using the monotonicity of $L_i$ and union bound, we have the following.

**Lemma 22.**

$$\mathbb{P}(\mathcal{M}_\varepsilon) \geq 1 - SH \exp\left(-\frac{1}{2} L_{i^*} \log \frac{1}{2e\delta}\right) - SHi_\varepsilon \exp\left(-\frac{1}{2} L_{i^*} \log \frac{SAH}{e\delta_{\text{samp}}}\right)$$

$$= 1 - \exp\left(-\tilde{\Theta}\left(\frac{\varepsilon B}{C_{\text{L2E}}(B)}\right)\right).$$

Let $W_h(S) \in (2^{-i}, 2^{-i+1}]$ and assume that $\mathcal{D}_{1,\delta}$ happened for at least $\frac{L_j}{2}$, where $j$ is the index that $(s, a)$ is filtered. We denote the number of $(s, a)$ samples at horizon $h$ when running each policy in a policy set $\Pi$ $A$ times as $N_\Pi^A(s, a, h)$. Let $I \subset [L_j]$ be the set of indices that $\mathcal{D}_{1,\delta}$ happened, which means $|I| \geq L_j/2$. Assume $m \in I$. If we rerun each policy in $\Pi_{j,m}$ once,

$$\mathbb{P}(\# \text{ of } (s, a) \text{ samples at horizon } h < \frac{1}{4} N_j) \leq \frac{\delta_{\text{samp}}}{H}$$

by Lemma 18. Now consider rerunning each policy in $\Pi_{j,m}$ $A$ times. Since running policies are independent, we can think of the process as $A$ repetition of running each policy in $\Pi_{j,m}$ once. Thus, we get

$$\mathbb{P}(N_{\Pi_{j,m}}^A(s, a, h) < \frac{1}{8} AN_j) \leq \mathbb{P}(\sum_{i=1}^{A} \mathbb{I}^i_{\{N_{\Pi_{j,m}}^1(s,a,h)<\frac{1}{4}N_j\}} \geq \frac{A}{2}) \leq \exp(-\frac{A}{2} \ln(H/2e\delta_{\text{samp}})),$$

where $\mathbb{I}^i$ is the indicator function for $i$-th repetition of running each policy in $\Pi_{j,m}$ and the second inequality follows from Lemma 21. If we rerun each policy in $\Pi_j$ $A$ times,

$$\mathbb{P}(N_{\Pi_j}^A < \frac{1}{32} AN_j L_j) \leq \mathbb{P}(\sum_{m \in I} \mathbb{I}_{\{N_{\Pi_{j,m}}^A(s,a,h)<\frac{1}{8}AN_j\}} \geq \frac{|I|}{2}) \leq \exp(-\frac{Y}{2} \ln(1/2e \exp(-\frac{A}{2} \ln(H/2e\delta_{\text{samp}}))))$$

$$\leq \exp(-\frac{L_j}{4} \ln(1/2e \exp(-\frac{A}{2} \ln(H/2e\delta_{\text{samp}}))))$$

$$\leq \exp\left(-\tilde{\Theta}\left(AL_j\right)\right)$$

by Lemma 21. If this happens, let's say that $(s, a)$ is *well-collected* at horizon $h$ for $A$ repetition. However, the failure probability

$$\mathbb{P}(\mathcal{D}_{1,\delta} \text{ happened less than } \frac{L_j}{2}) \leq \exp\left(-\tilde{\Theta}\left(L_j\right)\right),$$

which is more dominant. Thus, the following holds.

**Lemma 23.** *Consider $s$ whose reachable index at $h$ is $i \leq i_\varepsilon$. If we replay policies saved for $(s, a)$ $A$ times, the number $T_{hs}$ of $(s, a)$ samples we get satisfies*

$$\mathbb{P}\left(T_{hs} < \frac{AN_i L_i}{16}\right) \leq \exp\left(-\tilde{\Theta}\left(\frac{\varepsilon B}{C_{\text{L2E}}(B)}\right)\right).$$

## D    ANALYSIS OF SAR

Fix $\varepsilon \geq 0$. We say that an arm $i$ of a bandit $m$ is $\varepsilon$-*good* if $\mu_{m,1} - \mu_{m,i} \leq \varepsilon$. An arm is $\varepsilon$-*bad* if it is not $\varepsilon$-good. Let $g_m(\varepsilon)$ denote the number of $\varepsilon$-good arms in bandit $m$. We write $k^* := \max\left\{k : \bar{\Delta}_{(KM+1-k)} > \varepsilon\right\}$ and define the following two key events:

$$\mathcal{E}_1 = \left\{\forall k \in [k^*], \quad \frac{\varepsilon}{2}\text{-good pairs are not rejected at the end of phase } k\right\}$$

$$\mathcal{E}_2 = \{\forall k \in [(k^* + 1), \ldots, K], \quad \text{for every active bandit } m \text{ containing an } \varepsilon\text{-bad arm}$$

$$\text{at the beginning of phase } k, \text{ an } \tfrac{\varepsilon}{2}\text{-good arm in bandit } m \text{ is not rejected}\}$$

We first show that the intersection of these two events leads to a successful good arm identification for every bandit.

**Lemma 24.** *Suppose $\mathcal{E}_1 \cap \mathcal{E}_2$ holds. Then for every $m \in [M]$, the accepted arm is $\varepsilon$-good.*

*Proof.* Suppose the conclusion is not true; i.e., there exists a bandit $m$ for which an $\varepsilon$-bad arm $(m, b)$ has been accepted. Then, there exists a phase $k \in [KM - 1]$ where the best arm $(m, 1)$ is rejected from bandit $m$. Due to $\mathcal{E}_1$ and the fact that arm $(m, 1)$ is an $\tfrac{\varepsilon}{2}$-good arm, we know $k \geq k^* + 1$. Now, at the beginning of phase $k$, the bandit $m$ must contain both $(m, b)$ and $(m, 1)$. However, due to $\mathcal{E}_2$, the arm $(m, 1)$ cannot be rejected, which contradicts our supposition. $\square$

Furthermore, consider the following event

$$\mathcal{E}_0 = \left\{ \forall m \in [M], \forall i \in [K], \forall k \in [MK - 1], \left| \hat{\mu}_{m,i}(n_k) - \mu_{m,i} \right| < \frac{1}{8}(\bar{\Delta}_{(MK+1-k)} \vee \bar{\Delta}_{(g(\varepsilon)+1)}) \right\}$$

**Lemma 25.** $\mathcal{E}_0 \implies \mathcal{E}_1 \cap \mathcal{E}_2$

*Proof.* Assume $\mathcal{E}_0$. To show $\mathcal{E}_1$, it suffices to show that, for every $k \in [k^*]$, if no $\tfrac{\varepsilon}{2}$-good arm was rejected before phase $k$ then no $\tfrac{\varepsilon}{2}$-good arm will be rejected in phase $k$ (i.e., either accepts an arm or rejects a non-$\tfrac{\varepsilon}{2}$-good arm).

So, let $k \in [k^*]$, which implies that $\bar{\Delta}_{(MK+1-k)} > \varepsilon$ by definition, and assume that no $\tfrac{\varepsilon}{2}$-good arm was rejected before phase $k$. Furthermore, $\mathcal{E}_1$ is trivially true if the phase $k$ accepts an arm. Thus, it suffices to assume that the phase $k$ does not accept an arm.

We claim that, at the beginning of phase $k$, there exists an arm $(\bar{m}, \bar{i}) \in S$ such that

$$\mu_{\bar{m},1} - \mu_{\bar{m},\bar{i}} \geq \bar{\Delta}_{(MK+1-k)} \ .$$

Hereafter, we omit $(n_k)$ from $\hat{\mu}_{\cdot,\cdot}(n_k)$. To prove this claim, first note that there exists $(m', i') \in S$ such that

$$\bar{\Delta}_{m',i'} \geq \bar{\Delta}_{(MK+1-k)} \ .$$

(To see this, first, confirm that this is true with equality if the arm $(MK + 1 - k)$ is rejected or accepted at each phase $k$; now, notice that if an arm other than $(MK + 1 - k)$ was rejected or accepted, then it only makes the equality into $\geq$.) Then, we have the following two cases:

- If $i' \neq 1$, then $\bar{\Delta}_{m',i'} = \mu_{m',1} - \mu_{m',i'}$ by definition, so we can take $\bar{m} = m'$ and $\bar{i} = i'$ to prove the claim.

- If $i' = 1$, then, since phase $k$ does not accept an arm, there must exist another surviving arm $i'' \neq 1$ in bandit $m'$. Since $\bar{\Delta}_{m',i''} = \mu_{m',1} - \mu_{m',i''}$ and

$$\bar{\Delta}_{m',i''} \geq \bar{\Delta}_{m',2} = \bar{\Delta}_{m',1} = \bar{\Delta}_{m',i'} \geq \bar{\Delta}_{(MK+1-k)} \ ,$$

  we can choose $\bar{m} = m'$ and $\bar{i} = i''$ to prove the claim.

Assume that $\mathcal{E}_1$ is false; i.e., an $\tfrac{\varepsilon}{2}$-good arm in bandit $m$ is rejected. This implies that there exists an active bandit $m$ such that

$$\exists g \in [g_m(\tfrac{\varepsilon}{2})] : \hat{\mu}_{m,\hat{1}_m} - \hat{\mu}_{m,g} \geq \hat{\mu}_{\bar{m},\hat{1}_{\bar{m}}} - \hat{\mu}_{\bar{m},\bar{i}} \ .$$

Note that, using $\mathcal{E}_0$ and $\mu_{m,\hat{1}_m} - \mu_{m,g} \leq \mu_{m,1} - \mu_{m,g} \leq \tfrac{\varepsilon}{2} < \tfrac{1}{2}\bar{\Delta}_{(MK+1-k)}$,

$$(\text{LHS}) = \hat{\mu}_{m,\hat{1}_m} - \mu_{m,\hat{1}_m} + \mu_{m,\hat{1}_m} - \mu_{m,g} + \mu_{m,g} - \hat{\mu}_{m,g}$$

$$< \frac{\bar{\Delta}_{(MK+1-k)}}{8} + \frac{\bar{\Delta}_{(MK+1-k)}}{2} + \frac{\bar{\Delta}_{(MK+1-k)}}{8}$$

$$= \frac{3}{4}\bar{\Delta}_{(MK+1-k)} \ .$$

On the other hand,

$$
\begin{aligned}
(\text{RHS}) &\geq \hat{\mu}_{\bar{m},1} - \hat{\mu}_{\bar{m},\bar{i}} && ((m,1) \in S \text{ since no } \tfrac{\varepsilon}{2}\text{-good arm rejected before phase } k) \\
&= \hat{\mu}_{\bar{m},1} - \mu_{\bar{m},1} + \mu_{\bar{m},1} - \mu_{\bar{m},\bar{i}} + \mu_{\bar{m},\bar{i}} - \hat{\mu}_{\bar{m},\bar{i}} \\
&> -\frac{1}{8}\bar{\Delta}_{(MK+1-k)} + \bar{\Delta}_{(MK+1-k)} - \frac{1}{8}\bar{\Delta}_{(MK+1-k)} \\
&\geq \frac{3}{4}\bar{\Delta}_{(MK+1-k)} \ .
\end{aligned}
$$

This is a contradiction.

We now prove $\mathcal{E}_2$. Suppose not; there exists a phase $k \geq k^* + 1$ and a bandit $m$ active at the beginning of phase $k$ where an $\frac{\varepsilon}{2}$-good arm $(g,m)$ is rejected even if there was a surviving bad arm $(b,m)$. This means that

$$\hat{\mu}_{m,g} \leq \hat{\mu}_{m,b}$$

On the other hand, note that $k \geq k^* + 1$ implies $\bar{\Delta}_{(MK+1-k)} \leq \bar{\Delta}_{(g(\varepsilon)+1)}$, so $\bar{\Delta}_{(MK+1-k)} \vee \bar{\Delta}_{(g(\varepsilon)+1)} = \bar{\Delta}_{(g(\varepsilon)+1)}$. Thus,

$$
\begin{aligned}
\hat{\mu}_{m,g} - \hat{\mu}_{m,b} &= \hat{\mu}_{m,g} - \mu_{m,g} + \mu_{m,g} - \mu_{m,b} + \mu_{m,b} - \hat{\mu}_{m,b} \\
&> -\frac{1}{8}\bar{\Delta}_{(g(\varepsilon)+1)} + \mu_{m,g} - \mu_{m,b} - \frac{1}{8}\bar{\Delta}_{(g(\varepsilon)+1)} && (\mathcal{E}_0) \\
&\geq -\frac{1}{8}\bar{\Delta}_{(g(\varepsilon)+1)} + \frac{1}{2}\bar{\Delta}_{(g(\varepsilon)+1)} - \frac{1}{8}\bar{\Delta}_{(g(\varepsilon)+1)} && (\text{definition of } g \text{ and } b) \\
&> 0
\end{aligned}
$$

This is a contradiction. $\qquad\square$

Let

$$H_1(\varepsilon) := \sum_{i=1}^{MK} \frac{1}{(\bar{\Delta}_{(i)} \vee \varepsilon)^2}, \quad H_2(\varepsilon) := \max_{i \geq g(\varepsilon)+1} \frac{i}{\bar{\Delta}_{(i)}^2}.$$

We present a relation between these two gap-dependent quantities.

**Lemma 26.** $H_2(\varepsilon) \leq H_1(\varepsilon) \leq \frac{g(\varepsilon)}{\varepsilon^2} + \log(\frac{MK}{g(\varepsilon)})H_2(\varepsilon)$.

*Proof.* Let $i^* = \arg\max_{i \geq g(\varepsilon)+1} i\bar{\Delta}_i^{-2}$. Note that

$$
\begin{aligned}
H_1(\varepsilon) = \sum_{i \geq 1} (\bar{\Delta}_i \vee \varepsilon)^{-2} &\geq \sum_{i=1}^{g(\varepsilon)} \bar{\Delta}_{g(\varepsilon)+1}^{-2} + \sum_{i \geq g(\varepsilon)+1} \Delta_i^{-2} \\
&\geq \sum_{i=1}^{g(\varepsilon)} \bar{\Delta}_{g(\varepsilon)+1}^{-2} + \sum_{i=g(\varepsilon)+1}^{i^*} \bar{\Delta}_{i^*}^{-2} \\
&= \sum_{i=1}^{g(\varepsilon)} \bar{\Delta}_{g(\varepsilon)+1}^{-2} + (i^* - g(\varepsilon))\bar{\Delta}_{i^*}^{-2} \\
&= \sum_{i=1}^{g(\varepsilon)} \bar{\Delta}_{g(\varepsilon)+1}^{-2} + H_2(\varepsilon) - g(\varepsilon)\bar{\Delta}_{i^*}^{-2} \\
&\geq \sum_{i=1}^{g(\varepsilon)} \bar{\Delta}_{g(\varepsilon)+1}^{-2} + H_2(\varepsilon) - g(\varepsilon)\bar{\Delta}_{g(\varepsilon)+1}^{-2} \\
&\geq H_2(\varepsilon).
\end{aligned}
$$

For the right inequality,

$$H_1(\varepsilon) = \sum_{i \geq 1} \frac{1}{i} i(\bar{\Delta}_i \vee \varepsilon)^{-2} =$$

$$\leq \sum_{i=1}^{g(\varepsilon)} \frac{1}{i} i \varepsilon^{-2} + \sum_{i=g(\varepsilon)+1}^{MK} \frac{1}{i} H_2(\varepsilon)$$

$$\leq \frac{g(\varepsilon)}{\varepsilon^2} + \log(\frac{MK}{g(\varepsilon)}) H_2(\varepsilon).$$

$\square$

We are now ready to prove Theorem 3.2.

**Theorem D.1** (Refinement of Theorem 3.2). If we run Algorithm 2 with $B \geq MK$, then the total number of budget used is at most $B$ and

$$\mathbb{P}(\exists m \in [M] : \mu_{m,1} - \mu_{m,J_B(m)} > \varepsilon) \leq 2M^2 K^2 \exp\left(-\frac{B - MK}{128\sigma^2 \overline{\log}(MK) \cdot \max_{i \geq g(\varepsilon)+1} i \bar{\Delta}_{(i)}^{-2}}\right)$$

$$\leq 2M^2 K^2 \exp\left(-\frac{B - MK}{128\sigma^2 \overline{\log}(MK) \cdot \sum_{i \in [MK]} (\bar{\Delta}_{(i)} \vee \varepsilon)^{-2}}\right).$$

*Proof.* For the first part, the total budget used is bounded as

$$\sum_{k=1}^{MK-1} n_k(B, M, K) + n_{MK-1}(B, M, K) \leq MK + \frac{B - MK}{\overline{\log}(MK)} \left(\frac{1}{2} + \sum_{k=1}^{MK-1} \frac{1}{MK + 1 - k}\right) = B,$$

where we used $\lceil x \rceil \leq 1 + x$ For the second part, it suffices to bound $\mathbb{P}(\bar{\mathcal{E}}_0)$ by Lemma 24 and Lemma 25. Fix a bandit $m$ and an arm $i$. Then,

$$\mathbb{P}\left(\exists k \in [KM - 1] : \left|\hat{\mu}_{m,i}(n_k) - \mu_{m,i}\right| \geq \frac{1}{8}(\bar{\Delta}_{(MK+1-k)} \vee \bar{\Delta}_{(g(\varepsilon)+1)})\right)$$

$$\leq \sum_{k=1}^{KM-1} 2 \exp\left(-\frac{n_k}{2\sigma^2} \cdot \frac{(\bar{\Delta}_{(MK+1-k)} \vee \bar{\Delta}_{(g(\varepsilon)+1)})^2}{64}\right)$$

$$\leq \sum_{k=1}^{KM-1} 2 \exp\left(-\frac{B - MK}{\overline{\log}(MK) \cdot (MK + 1 - k)} \frac{(\bar{\Delta}_{(MK+1-k)} \vee \bar{\Delta}_{(g(\varepsilon)+1)})^2}{128\sigma^2}\right)$$

$$\leq 2MK \exp\left(-\frac{B - MK}{128\sigma^2 \overline{\log}(MK) \cdot \max_{i \in [2..MK]} i(\bar{\Delta}_{(i)} \vee \bar{\Delta}_{(g(\varepsilon)+1)})^{-2}}\right)$$

$$\leq 2MK \exp\left(-\frac{B - MK}{128\sigma^2 \overline{\log}(MK) \cdot \max_{i \geq g(\varepsilon)+1} i \bar{\Delta}_{(i)}^{-2}}\right).$$

Taking a union bound over $m \in [M]$ and $i \in [K]$ and Lemma 26 completes the proof. $\square$

Note that when $\varepsilon = 0$, this theorem recovers the best arm identification result of Bubeck et al. (2013).

**Remark 27.** *If we set $M = 1$, SAR becomes a single bandit algorithm. Consider running this single bandit SAR to each bandit $m \in [M]$ with budget $B/M$. Then, we have*

$$\mathbb{P}(\mu_{m,1} - \mu_{m,J(m)} > \varepsilon) \leq \exp\left(-\tilde{\Theta}\left(\frac{B/M}{\sum_i (\bar{\Delta}_{m,i} \vee \varepsilon)^{-2})}\right)\right)$$

*for each bandit $m \in [M]$. This yields*

$$\mathbb{P}(\exists m \in [M] : mu_{m,1} - \mu_{m,J(m)} > \varepsilon) \leq \exp\left(-\tilde{\Theta}\left(\frac{B/M}{\max_m \sum_i (\bar{\Delta}_{m,i} \vee \varepsilon)^{-2})}\right)\right),$$

*which is worse than the result of Theorem 3.2 since*

$$\sum_m \sum_i (\bar{\Delta}_{m,i} \vee \varepsilon)^{-2}) \leq M \max_m \sum_i (\bar{\Delta}_{m,i} \vee \varepsilon)^{-2}).$$

*Due to this difference, if we use single bandit algorithm in BREA, we get the term*

$$\exp\left(-\tilde{\Theta}\left(\frac{B}{H^5 \max_{h \in [H]} C_h^2 S \max_{s \in \mathcal{S}} W_h(s)^{-1} \sum_{a \in \mathcal{A}} (\bar{\Delta}_h(s,a) \vee \frac{\varepsilon}{W_h(s)})^{-2}}\right)\right),$$

*which is worse than the actual term*

$$\exp\left(-\tilde{\Theta}\left(\frac{B}{H^5 \max_{h \in [H]} C_h^2 \sum_{s \in \mathcal{S}} W_h(s)^{-1} \sum_{a \in \mathcal{A}} (\bar{\Delta}_h(s,a) \vee \frac{\varepsilon}{W_h(s)})^{-2}}\right)\right)$$

*of Theorem 3.3.*

## E   PROOF OF THEOREM 3.3 AND COROLLARY 3

In this section, we provide an analysis of BREA. Recall that $\varepsilon \geq 2SH^2\varepsilon_B$ and $i_\varepsilon = \lceil \log_2(\frac{2SH^2}{\varepsilon}) \rceil$
We define the events

$$\mathcal{M}_{h,\varepsilon} = \Big\{ \text{For any } s \in \mathcal{S},$$

$$\text{FB-L2E}(\{(s,1)\}, h, B') \text{ outputs } \mathcal{X}_i = \{(s,1)\} \text{ for some } i \leq i_\varepsilon \implies 2^{-i-3} \leq W_h(s) \leq 2^{-i+1},$$

$$\text{FB-L2E}(\{(s,1)\}, h, B') \text{ outputs } \mathcal{X}_i = \emptyset \text{ for all } i \in [i_\varepsilon] \implies W_h(s) \leq \frac{\varepsilon}{2SH^2} \Big\},$$

$$\mathcal{M}_\varepsilon = \cup_{h=1}^H \mathcal{M}_{h,\varepsilon},$$

$$\mathcal{L}_{h,\varepsilon} = \Big\{ \text{For any } i \leq i_\varepsilon \text{ and any phase } k \in [|\mathcal{Z}_{hi}|A - 1],$$

$$\text{each } (s,a) \in A_k \text{ is collected at least } \lfloor \frac{n_k}{N_i^{sh}} \rfloor \frac{N_i^{sh}}{8} \text{ times} \Big\},$$

$$\mathcal{L}_\varepsilon = \cup_{h=1}^H \mathcal{L}_{h,\varepsilon}$$

$$\mathcal{E}_h = \Big\{ \Delta_h^{\hat{\pi}}(s, \hat{\pi}_h(s)) \leq \frac{\varepsilon}{2C_h H W_h(s)} \text{ for all } s \in \cup_{i=1}^{i_\varepsilon} \mathcal{Z}_{hi} \Big\}.$$

$$(8)$$

Before proving Theorem 3.3, we provide lemmas that will give us a relation between the suboptimality gap and its empirical estimate.

**Lemma 28.** *Let $0 < a \leq b$ and assume $f_1, f_2 \geq 0$ satisfy $|f_1 - f_2| \leq b$. Then*

$$(f_1 \vee a)^{-2} \leq (\frac{a}{2b} f_2 \vee a)^{-2}.$$

*Proof.* If $f_1 \leq a$, then $(f_1 \vee a)^{-2} = a^{-2}$. On the other hand, $f_2 \leq f_1 + b \leq a + b \leq 2b$. Thus,

$$(f_1 \vee a)^{-2} = a^{-2} = (\frac{a}{2b} f_2 \vee a)^{-2}.$$

If $f_1 > a$, then $(f_1 \vee a)^{-2} = f_1^{-2} < a^{-2}$. Also, $f_2 \leq f_1 + b < f_1 + \frac{f_1}{a}b = f_1(1 + \frac{b}{a}) \leq \frac{2b}{a}f_1$. Thus,

$$(f_1 \vee a)^{-2} = f_1^{-2} < (\frac{a}{2b} f_2 \vee a)^{-2}.$$

$\square$

**Lemma 29.** *On $\cap_{h'=H}^{h+1} \mathcal{E}_{h'} \cap \mathcal{M}_\varepsilon \cap \mathcal{L}_\varepsilon$, we have*

$$(\Delta_h^{\hat{\pi}}(s,a) \vee \frac{\varepsilon}{2C_h H W_h(s)})^{-2} \leq 16 C_h^2 H^2 (\Delta_h(s,a) \vee \frac{2\varepsilon}{W_h(s)})^{-2}.$$

*Proof.* By Lemma 7, for any policy $\pi'$

$$
\sum_s w_{h+1}^{\pi'}(s)(V_{h+1}^*(s) - V_{h+1}^{\hat\pi}(s)) \leq \sum_{h'=h+1}^{H} \sup_\pi \sum_s w_{h'}^\pi(s)\varepsilon_h(s)
$$

$$
\leq \sum_{h'=h+1}^{H} \sup_\pi \sum_{i \leq i_\varepsilon} \sum_{s \in \mathcal{Z}_{hi}} w_{h'}^\pi(s)\frac{\varepsilon}{2C_h HW_h(s)} + H\sum_{h'=h+1}^{H}\sum_{s \notin \cup_{i \leq i_\varepsilon}\mathcal{Z}_{hi}} \sup_\pi w_{h'}^\pi(s)
$$

$$
\leq \sum_{h'=h+1}^{H} \frac{\varepsilon}{2H} + \sum_{h'=h+1}^{H} SH\frac{\varepsilon}{2SH^2}
$$

$$
\leq \varepsilon.
$$

By Lemma 8,

$$
|\Delta_h(s,a) - \Delta_h^{\hat\pi}(s,a)| \leq \frac{\varepsilon}{W_h(s)}.
$$

By applying Lemma 28 with $f_1 = \Delta_h^{\hat\pi}(s,a), f_2 = \Delta_h(s,a), a = \frac{\varepsilon}{2C_h HW_h(s)}, b = \frac{\varepsilon}{W_h(s)}$, the proof is done. $\square$

**Theorem E.1** (Theorem 3.3). *If we run Algorithm 3 with*

$$
B \geq \max\{2SHc(\frac{B}{2SH}), 2SA\varepsilon_{\frac{B}{2SH}}\log_2\frac{1}{\varepsilon_{\frac{B}{2SH}}}\},
$$

*then the total number of budget used is at most $B$. Moreover, for any $\varepsilon \geq 2SH^2\varepsilon_{\frac{B}{2SH}}$,*

$$
\mathbb{P}\left(V_0^* - V_0^{\hat\pi} > \varepsilon\right) \leq \exp\left(-\tilde\Theta\left(\frac{\varepsilon B}{C_{\mathrm{L2E}}(\frac{B}{2SH})}\right)\right)
$$

$$
+ \exp\left(-\tilde\Theta\left(\frac{B}{H^5\max_{h\in[H]}C_h^2\sum_{s\in\mathcal{S}}W_h(s)^{-1}\sum_{a\in\mathcal{A}}(\bar\Delta_h(s,a)\vee\frac{\varepsilon}{W_h(s)})^{-2}}\right)\right).
$$

*Proof.* The budget used from the first part is

$$
SH\lfloor\frac{B}{2SH}\rfloor \leq \frac{B}{2}
$$

by Theorem 3.1. For the second part, we use

$$
\sum_{i=1}^{|\mathcal{Z}_{hi}|A-1} T_i(s,a) + T_{|\mathcal{Z}_{hi}|A-1}(s,a)
$$

$$
\leq \frac{1}{N_i}\left(\sum_{i=1}^{|\mathcal{Z}_{hi}|A-1} n_i + n_{|\mathcal{Z}_{hi}|A-1}\right)
$$

$$
\leq \frac{\lfloor B''2^{-i-2}\rfloor}{2^{i+2}} \leq B'' = \frac{B}{2HJ} \tag{Theorem 3.2}
$$

for each multiple bandit $\mathcal{Z}_{hi}$. Thus, the budget used in the second part is at most $\frac{B}{2}$, the total budget used is at most $B$.

We now prove the probability bound. By Theorem 3.1 and that $B \geq 2SHc(\frac{B}{2SH})$, we have

$$
\mathbb{P}(\mathcal{M}_\varepsilon^\mathsf{c}) \leq SH\exp\left(-\tilde\Theta\left(\frac{\varepsilon B}{C_{\mathrm{L2E}}(\frac{B}{2SH})}\right)\right) = \exp\left(-\tilde\Theta\left(\frac{\varepsilon B}{C_{\mathrm{L2E}}(\frac{B}{2SH})}\right)\right),
$$

$$
\mathbb{P}(\mathcal{L}_\varepsilon^\mathsf{c}) \leq S^2A^2H\exp\left(-\tilde\Theta\left(\frac{\varepsilon B}{C_{\mathrm{L2E}}(\frac{B}{2SH})}\right)\right) = \exp\left(-\tilde\Theta\left(\frac{\varepsilon B}{C_{\mathrm{L2E}}(\frac{B}{2SH})}\right)\right). \tag{9}
$$

We can decompose the probability as

$$\mathbb{P}(V_0^* - V_0^{\hat{\pi}} > \varepsilon) \le \mathbb{P}(V_0^* - V_0^{\hat{\pi}} > \varepsilon, \mathcal{M}_\varepsilon, \mathcal{L}_\varepsilon) + \mathbb{P}(\mathcal{M}_\varepsilon^{\mathsf{c}}) + \mathbb{P}(\mathcal{L}_\varepsilon^{\mathsf{c}})$$

$$\le \mathbb{P}(V_0^* - V_0^{\hat{\pi}} > \varepsilon, \mathcal{M}_\varepsilon, \mathcal{L}_\varepsilon) + \exp\left(-\tilde{\Theta}\left(\frac{\varepsilon B}{C_{\mathrm{L2E}}(\frac{B}{2SH})}\right)\right). \quad (10)$$

Assume that $\mathcal{M}_\varepsilon, \mathcal{L}_\varepsilon, \{\mathcal{E}_h\}_{h=1}^H$ holds. Then, by Lemma 7,

$$V_0^* - V_0^{\hat{\pi}} \le \sum_{h=1}^H \sup_\pi \sum_s w_h^\pi(s)\varepsilon_h(s)$$

$$\le \sum_{h=1}^H \sup_\pi \sum_{i \le i_\varepsilon} \sum_{s \in \mathcal{Z}_{hi}} w_h^\pi(s)\frac{\varepsilon}{2C_h H W_h(s)} + H\sum_{h=1}^H \sum_{s \notin \cup_{i \le i_\varepsilon} \mathcal{Z}_{hi}} \sup_\pi w_h^\pi(s)$$

$$\le \sum_{h=1}^H \frac{\varepsilon}{2H} + SH^2 \frac{\varepsilon}{2SH^2}$$

$$\le \frac{\varepsilon}{2} + \frac{\varepsilon}{2} = \varepsilon,$$

where the second inequality follows from the definition of $C_h$. Thus, we have

$$\mathbb{P}(V_0^* - V_0^{\hat{\pi}} > \varepsilon, \mathcal{M}_\varepsilon, \mathcal{L}_\varepsilon) \le \sum_{h=1}^H \mathbb{P}(\mathcal{E}_h^{\mathsf{c}}, \mathcal{M}_\varepsilon, \mathcal{L}_\varepsilon, \cap_{h'=H}^{h+1}\mathcal{E}_{h'}). \quad (11)$$

We try to bound $\mathbb{P}(\mathcal{E}_h^{\mathsf{c}}, \mathcal{M}_\varepsilon, \mathcal{L}_\varepsilon, \cap_{h'=H}^{h+1}\mathcal{E}_{h'})$.

On the event $\mathcal{L}_\varepsilon$, every multiple bandit instance $\mathcal{Z}_{hi}$ effectively collects samples so that SAR with budget $\Theta(\frac{B}{2HJ}2^{-i-2})$ is run. On the event $\mathcal{M}_\varepsilon$, this is $\Theta(\frac{BW_h(s)}{HJ}) = \Theta(\frac{BW_h(s)}{H})$. To be precise, the minimum budget of SAR is $\min_{s \in \mathcal{Z}_h} \frac{BW_h(s)}{2HJ} \ge \frac{B\varepsilon \frac{B}{2SH}}{2HJ}$ and this is more or equal to $SA$ by the hypothesis of Theorem 3.3. Thus, by Theorem 3.2, we have

$$\mathbb{P}\left(\Delta_h^{\hat{\pi}}(s, \hat{\pi}_h(s)) > \frac{\varepsilon}{2C_h H W_h(s)} \text{ for some } s \in \mathcal{Z}_{hi}, \mathcal{M}_\varepsilon, \mathcal{L}_\varepsilon, \cap_{h'=H}^{h+1}\mathcal{E}_{h'}|\mathcal{F}_{h+1}\right)$$

$$\le \exp\left(-\tilde{\Theta}\left(\frac{B}{H^3 \sum_{(s,a)\in\mathcal{Z}_{hi}\times\mathcal{A}} W_h(s)^{-1}(\bar{\Delta}_h^{\hat{\pi}}(s,a) \vee \frac{\varepsilon}{2C_h H W_h(s)})^{-2}}\right)\right)$$

$$= \exp\left(-\tilde{\Theta}\left(\frac{B}{H^3 \sum_{s\in\mathcal{Z}_{hi}} W_h(s)^{-1}\sum_{a\ge 2}(\Delta_h^{\hat{\pi}}(s,a) \vee \frac{\varepsilon}{2C_h H W_h(s)})^{-2}}\right)\right)$$

$$\le \exp\left(-\tilde{\Theta}\left(\frac{B}{C_h^2 H^5 \sum_{s\in\mathcal{Z}_{hi}} W_h(s)^{-1}\sum_{a\ge 2}(\Delta_h(s,a) \vee \frac{\varepsilon}{W_h(s)})^{-2}}\right)\right)$$

$$\le \exp\left(-\tilde{\Theta}\left(\frac{B}{C_h^2 H^5 \sum_{(s,a)\in\mathcal{Z}_{hi}\times\mathcal{A}} W_h(s)^{-1}(\bar{\Delta}_h(s,a) \vee \frac{\varepsilon}{W_h(s)})^{-2}}\right)\right), \quad (12)$$

where the second inequality follows from Lemma 29, $\mathcal{F}_{h+1}$ is a filtration up to learning in step $h+1$. Thus, we have

$$\mathbb{P}(\mathcal{E}_h^{\mathsf{c}}, \mathcal{M}_\varepsilon, \mathcal{L}_\varepsilon, \cap_{h'=H}^{h+1}\mathcal{E}_{h'}) \le i_\varepsilon \exp\left(-\tilde{\Theta}\left(\frac{B}{C_h^2 H^5 \sum_{(s,a)\in\mathcal{S}\times\mathcal{A}} W_h(s)^{-1}(\Delta_h(s,a) \vee \frac{\varepsilon}{W_h(s)})^{-2}}\right)\right)$$

$$= \exp\left(-\tilde{\Theta}\left(\frac{B}{C_h^2 H^5 \sum_{(s,a)\in\mathcal{S}\times\mathcal{A}} W_h(s)^{-1}(\Delta_h(s,a) \vee \frac{\varepsilon}{W_h(s)})^{-2}}\right)\right).$$

If we plug this into equation 11 and equation 10, we get the probability bound of the theorem. $\square$

**Corollary 30** (Exact statement of Corollary 3). *If*

$$2SH^2\varepsilon_{\frac{B}{2SH}} < \varepsilon^* := \min\{\min_{s,h}^+ W_h(s), 2H \min_{s,a,h}^+ C_h W_h(s)\bar{\Delta}_h(s,a)\},$$

*we obtain a guarantee of the best policy identification, given by*

$$\mathbb{P}\left(V_0^* - V_0^{\hat{\pi}} > 0\right) \leq \exp\left(-\tilde{\Theta}\left(\frac{\varepsilon^* B}{C_{\mathrm{L2E}}(\frac{B}{2SH})}\right)\right)$$

$$+ \exp\left(-\tilde{\Theta}\left(\frac{B}{H^5 \max_{h\in[H]} C_h^2 \sum_{s\in\mathcal{S}} W_h(s)^{-1} \sum_{a\in\mathcal{A}} (\bar{\Delta}_h(s,a) \vee \frac{\varepsilon^*}{W_h(s)})^{-2}}\right)\right).$$

*Furthermore, if the optimal action in each state $s$ at each step $h$ is unique, then*

$$\mathbb{P}\left(V_0^* - V_0^{\hat{\pi}} > 0\right) \leq \exp\left(-\tilde{\Theta}\left(\frac{\varepsilon^* B}{C_{\mathrm{L2E}}(\frac{B}{2SH})}\right)\right)$$

$$+ \exp\left(-\tilde{\Theta}\left(\frac{B}{H^3 \max_{h\in[H]} \sum_{s\in\mathcal{S}} W_h(s)^{-1} \sum_{a\in\mathcal{A}} \bar{\Delta}_h(s,a)^{-2}}\right)\right).$$

*Proof.* Let's take any $\varepsilon_{\frac{B}{2SH}} \leq \varepsilon < \varepsilon^*$ and assume that the events $\mathcal{M}_\varepsilon, \mathcal{L}_\varepsilon, \cap_{h=1}^H \mathcal{E}_h$ hold. By the definition of $\mathcal{M}_\varepsilon$ in equation 8, $\varepsilon < 2SH^2 \min_{s,h}^+ W_h(s)$ implies that any state $s$ with $W_h(s) > 0$ lies in $\mathcal{Z}_h$ for any $h \in [H]$. Since $\Delta_H^{\hat{\pi}}(s,a) = \Delta_H(s,a)$, the event $\mathcal{E}_H$ and that $\varepsilon < 2H \min_{s,a,h}^+ C_h W_h(s)\Delta_h(\bar{s},a)$ implies that $\hat{\pi}_H(s)$ is an optimal action for all $s \in \mathcal{Z}_H$. Then this implies that $\Delta_{H-1}^{\hat{\pi}}(s,a) = \Delta_{H-1}(s,a)$ holds for all $s,a$. Again, the event $\mathcal{E}_{H-1}$ and that $\varepsilon < 2H \min_{s,a,h}^+ C_h W_h(s)\Delta_h(\bar{s},a)$ implies that $\hat{\pi}_{H-1}(s)$ is an optimal action for all $s \in \mathcal{Z}_{H-1}$. Repeating this procedure, we can conclude that $\hat{\pi}$ is optimal. The first probability bound follows by limiting the result of Theorem 3.3 as $\varepsilon \to \varepsilon^* -$.

Next, we further assume the uniqueness of the optimal actions. In equation 12 of the proof of Theorem 3.3, we may apply tha fact that $\Delta_h^{\hat{\pi}}(s,a) = \Delta_h(s,a)$ and $\varepsilon < 2H \min_{s,a,h}^+ C_h W_h(s)\Delta_h(\bar{s},a)$ instead of Lemma 29 so that we obtain

$$\mathbb{P}\left(\Delta_h^{\hat{\pi}}(s, \hat{\pi}_h(s)) > \frac{\varepsilon}{2C_h H W_h(s)} \text{ for some } s \in \mathcal{Z}_{hi}, \mathcal{M}_\varepsilon, \mathcal{L}_\varepsilon, \cap_{h'=H}^{h+1} \mathcal{E}_{h'} | \mathcal{F}_{h+1}\right)$$

$$\leq \exp\left(-\tilde{\Theta}\left(\frac{B}{H^3 \sum_{(s,a)\in\mathcal{Z}_{hi}\times\mathcal{A}} W_h(s)^{-1}(\bar{\Delta}_h^{\hat{\pi}}(s,a) \vee \frac{\varepsilon}{2C_h H W_h(s)})^{-2}}\right)\right)$$

$$= \exp\left(-\tilde{\Theta}\left(\frac{B}{H^3 \sum_{s\in\mathcal{Z}_{hi}} W_h(s)^{-1} \sum_{a\geq 2}(\Delta_h^{\hat{\pi}}(s,a) \vee \frac{\varepsilon}{2C_h H W_h(s)})^{-2}}\right)\right)$$

$$= \exp\left(-\tilde{\Theta}\left(\frac{B}{H^3 \sum_{s\in\mathcal{Z}_{hi}} W_h(s)^{-1} \sum_{a\geq 2}(\Delta_h(s,a) \vee \frac{\varepsilon}{2C_h H W_h(s)})^{-2}}\right)\right)$$

$$= \exp\left(-\tilde{\Theta}\left(\frac{B}{H^3 \sum_{s\in\mathcal{Z}_{hi}} W_h(s)^{-1} \sum_{a\geq 2}\Delta_h(s,a)^{-2}}\right)\right)$$

$$\leq \exp\left(-\tilde{\Theta}\left(\frac{B}{H^3 \sum_{s\in\mathcal{S}} W_h(s)^{-1} \sum_{a\in\mathcal{A}} \bar{\Delta}_h(s,a)^{-2}}\right)\right).$$

Union bound over $h$, combining with the exploration term and taking the limit as $\varepsilon \to \varepsilon^* -$ give the second probability bound. $\qquad\square$

## F  PROOF OF THEOREM 3.4

We present a modified algorithm, BREAP in Algorithm 5

BREAP additionally refine the policy. Intuitively, BREAP gathers good arms from the second part, additionally collects samples of them, and picks the empirically best actions.

Consider the situation where we run SAR with the set $\mathcal{Z}_{hi} \times \mathcal{A}$. Let

$$\hat{\Delta}_h^{\hat{\pi}}(s, a) := \max_{a'} \hat{Q}_h^{\hat{\pi}}(s, a') - \hat{Q}_h^{\hat{\pi}}(s, a)$$

and

$$\hat{g}_{hi}^{\hat{\pi}}(\varepsilon) := |\{(s, a) \in \mathcal{Z}_{hi} \times \mathcal{A} : \hat{\Delta}_h^{\hat{\pi}}(s, a) \leq \varepsilon\}|.$$

We define the set $\widehat{\mathrm{OPT}}_h(\varepsilon)$ as the last $\hat{g}_{hi}^{\hat{\pi}}(\frac{\varepsilon}{\hat{W}_h(s)})$ surviving pairs.

Let $k^*$ be

$$\max\{k : \bar{\Delta}_{(|\mathcal{Z}_{hi}|A+1-k)} < \varepsilon\},$$

where $\bar{\Delta}_{(1)} \geq \bar{\Delta}_{(2)} \geq \ldots \geq \bar{\Delta}_{(|\mathcal{Z}_{hi}|A)}$ and define the events

$$\mathcal{G}_{0,\varepsilon}(\mathcal{Z}_{hi}) = \{\forall k, \forall (s, a) \in \mathcal{Z}_{hi} \times \mathcal{A}, \quad |\hat{Q}_h^{\hat{\pi}}(s, a, n_k) - Q_h^{\hat{\pi}}(s, a)| < \frac{1}{8}(\bar{\Delta}_{(|\mathcal{Z}_{hi}|A+1-k)} \vee \frac{\varepsilon}{W_h(s)})\}$$

$$\mathcal{G}_{1,\varepsilon}(\mathcal{Z}_{hi}) = \{\forall k \in [k^*], \quad \frac{\varepsilon}{2W_h(s)} - \text{good pairs are not rejected at phase } k\}$$

$$\mathcal{G}_{2,\varepsilon}(\mathcal{Z}_{hi}) = \{\forall k > k^*, \quad \frac{\varepsilon}{2W_h(s)} - \text{good pairs are not rejected at phase } k \text{ if there exists a bad pair in the same state}\}$$

as in Appendix D. We omit $\mathcal{Z}_{hi}$ when there is no confusion. We also redefine the events

$$\mathcal{E}_h = \{\sup_{\pi} \sum_{s \in \mathcal{Z}_{h1:i_\varepsilon}} w_h^\pi(s) \Delta_h^{\hat{\pi}}(s, \hat{\pi}(s)) \leq \frac{\varepsilon}{2H}\},$$

where $\mathcal{Z}_{h1:i_\varepsilon} = \cup_{i=1}^{i_\varepsilon} \mathcal{Z}_{hi}$.

We state some lemmas describing the properties of SAR process.

**Lemma 31.** *Under the events $\mathcal{M}_\varepsilon, \mathcal{L}_\varepsilon$, the event $\mathcal{G}_{0,\varepsilon}$ implies $\mathcal{G}_{1,\varepsilon}$ and $\mathcal{G}_{2,\varepsilon}$.*

*Proof.* This is just a restatement of Lemma 25. $\qquad\square$

**Lemma 32.** *Under the events $\mathcal{M}_\varepsilon, \mathcal{L}_\varepsilon$,*

$$\mathbb{P}(\mathcal{G}_{0,\varepsilon}^{\mathsf{c}}) \leq \exp\left(-\tilde{\Theta}\left(\frac{W_h(s)B}{H^3 \max_i i(\Delta_{(i)} \vee \frac{\varepsilon}{W_h(s)})^{-2}}\right)\right).$$

*Proof.* This is just a restatement of Theorem 3.2. $\qquad\square$

**Lemma 33.** *Under the events $\mathcal{M}_\varepsilon, \mathcal{L}_\varepsilon, \mathcal{G}_{0,\varepsilon}$, if a pair $(s, a)$ is rejected in a phase $k \in [k^*]$, then*

$$\Delta_h^{\hat{\pi}}(s, a) > \frac{1}{2}\bar{\Delta}_{(|\mathcal{Z}_{hi}|A+1-k)}.$$

*Proof.* There exists a pair $(s', a')$ in the remaining set at the beginning of phase $k$ such that

$$\Delta_h^{\hat{\pi}}(s', a') \geq \bar{\Delta}_{(|\mathcal{Z}_{hi}|A+1-k)}.$$

Since $(s, a)$ is eliminated in phase $k \leq k^*$,

$$\hat{\Delta}_h(s, a)_k \geq \hat{\Delta}_h(s', a')_k,$$

where the subscript $k$ is for the empirical gap until phase $k$. Let $a_s^* \in \arg\max_a \hat{Q}_h^{\hat{\pi}}(s, a)_k$. Then we have

$$\hat{\Delta}_h(s, a)_k = \hat{Q}_h^{\hat{\pi}}(s, \hat{a}_s)_k - \hat{Q}_h^{\hat{\pi}}(s, a)_k$$
$$= \hat{Q}_h^{\hat{\pi}}(s, \hat{a}_s)_k - Q_h^{\hat{\pi}}(s, \hat{a}_s) + Q_h^{\hat{\pi}}(s, \hat{a}_s) - Q_h^{\hat{\pi}}(s, a) + Q_h^{\hat{\pi}}(s, a) - \hat{Q}_h^{\hat{\pi}}(s, a)_k$$

$$< \frac{\bar{\Delta}_{(|\mathcal{Z}_{hi}|A+1-k)}}{8} + Q_h^{\hat{\pi}}(s, a_s^*) - Q_h^{\hat{\pi}}(s, a) + \frac{\bar{\Delta}_{(|\mathcal{Z}_{hi}|A+1-k)}}{8}$$

$$= \Delta_h^{\hat{\pi}}(s, a) + \frac{\bar{\Delta}_{(|\mathcal{Z}_{hi}|A+1-k)}}{4}$$

under $\mathcal{G}_0$. On the other hand,

$$\hat{\Delta}_h(s', a')_k = \hat{Q}_h^{\hat{\pi}}(s', \hat{a}_{s'})_k - \hat{Q}_h^{\hat{\pi}}(s', a')_k$$

$$\geq \hat{Q}_h^{\hat{\pi}}(s', a_{s'}^*)_k - \hat{Q}_h^{\hat{\pi}}(s', a')_k$$

$$= \hat{Q}_h^{\hat{\pi}}(s', a_{s'}^*)_k - Q_h^{\hat{\pi}}(s', a_{s'}^*) + Q_h^{\hat{\pi}}(s', a_{s'}^*) - Q_h^{\hat{\pi}}(s', a) + Q_h^{\hat{\pi}}(s', a) - \hat{Q}_h^{\hat{\pi}}(s', a')_k$$

$$> -\frac{\bar{\Delta}_{(|\mathcal{Z}_{hi}|A+1-k)}}{8} + \Delta_h^{\hat{\pi}}(s', a') - \frac{\bar{\Delta}_{(|\mathcal{Z}_{hi}|A+1-k)}}{8}$$

$$\geq -\frac{\bar{\Delta}_{(|\mathcal{Z}_{hi}|A+1-k)}}{8} + \bar{\Delta}_{(|\mathcal{Z}_{hi}|A+1-k)} - \frac{\bar{\Delta}_{(|\mathcal{Z}_{hi}|A+1-k)}}{8}$$

$$= \frac{3\bar{\Delta}_{(|\mathcal{Z}_{hi}|A+1-k)}}{4}$$

under $\mathcal{G}_0$. Then

$$\Delta_h^{\hat{\pi}}(s, a) + \frac{\bar{\Delta}_{(|\mathcal{Z}_{hi}|A+1-k)}}{4} > \hat{\Delta}_h(s, a)_k \geq \hat{\Delta}_h(s', a')_k > \frac{3\bar{\Delta}_{(|\mathcal{Z}_{hi}|A+1-k)}}{4},$$

which implies

$$\Delta_h^{\hat{\pi}}(s, a) > \frac{1}{2}\bar{\Delta}_{(|\mathcal{Z}_{hi}|A+1-k)}.$$

$\square$

**Lemma 34.** *Under the events $\mathcal{M}_\varepsilon, \mathcal{L}_\varepsilon, \mathcal{G}_{0,\varepsilon}$, if a pair $(s, a)$ is accepted in a phase $k \in [k^*]$, then*

$$\Delta_h^{\hat{\pi}}(s, a) = 0.$$

*Proof.* Since $(s, a)$ is accepted in the phase $k \in [k^*]$, $(s, a')$ for the other actions $a'$ are rejected befor phase $k$. By Lemma 33, $\Delta_h^{\hat{\pi}}(s, a') > 0$ for the other actions $a'$. Thus, $\Delta_h^{\hat{\pi}}(s, a) = 0$. $\square$

**Lemma 35.** *Under the events $\mathcal{M}_\varepsilon, \mathcal{L}_\varepsilon, \cap_{h'=h+1}^{H}\mathcal{E}_{h'}$, and $\mathcal{G}_{0,\varepsilon}$,*

$$\hat{\Delta}_h^{\hat{\pi}}(s, a) \leq \frac{\varepsilon}{W_h(s)} \implies \Delta_h(s, a) \leq \frac{3\varepsilon}{W_h(s)}$$

*Proof.* Assume that $\hat{\Delta}_h^{\hat{\pi}}(s, a) \leq \frac{\varepsilon}{W_h(s)}$. We have

$$\Delta_h(s, a) - \hat{\Delta}_h^{\hat{\pi}}(s, a) \leq \underbrace{\Delta_h(s, a) - \Delta_h^{\hat{\pi}}(s, a)}_{(I)} + \underbrace{\Delta_h^{\hat{\pi}}(s, a) - \hat{\Delta}_h^{\hat{\pi}}(s, a)}_{(II)}.$$

(I) is less than or equal to $\frac{\varepsilon}{W_h(s)}$ in the good event $\cap_{h'=h+1}^{H}\mathcal{E}_{h'}$ by Lemma 8. Thus, under $\cap_{h'=h+1}^{H}\mathcal{E}_{h'}$,

$$\Delta_h(s, a) \leq \frac{2\varepsilon}{W_h(s)} + (II). \tag{13}$$

Let $a_s^* \in \arg\max Q_h^{\hat{\pi}}(s, a)$. Then we have

$$(II) = \Delta_h^{\hat{\pi}}(s, a) - \hat{\Delta}_h^{\hat{\pi}}(s, a) = \max_a Q_h^{\hat{\pi}}(s, a) - \max_a \hat{Q}_h^{\hat{\pi}}(s, a) + \hat{Q}_h^{\hat{\pi}}(s, a) - Q_h^{\hat{\pi}}(s, a)$$

$$\leq \underbrace{Q_h^{\hat{\pi}}(s, a_s^*) - \hat{Q}_h^{\hat{\pi}}(s, a_s^*)}_{(III)} + \underbrace{\hat{Q}_h^{\hat{\pi}}(s, a) - Q_h^{\hat{\pi}}(s, a)}_{(IV)}.$$

(1) If $(s, a)$ is accepted, then

$$(II) = \Delta_h^{\hat{\pi}}(s, a) - \hat{\Delta}_h^{\hat{\pi}}(s, a) \leq \Delta_h^{\hat{\pi}}(s, a) \leq \frac{\varepsilon}{W_h(s)}$$

by Theorem 3.2. Thus, $\Delta_h(s, a) \leq \frac{3\varepsilon}{W_h(s)}$.

(2) If $(s, a)$ is rejected in some phase $k > k^*$, then

$$(\text{II}) = \Delta_h^{\hat{\pi}}(s, a) - \hat{\Delta}_h^{\hat{\pi}}(s, a) \leq \Delta_h^{\hat{\pi}}(s, a) \leq \frac{\varepsilon}{W_h(s)}$$

since

$$\mathcal{G}_{0,\varepsilon} \implies \mathcal{G}_{0,2\varepsilon} \implies \mathcal{G}_{1,2\varepsilon}, \quad \mathcal{G}_{2,2\varepsilon}$$

and $\mathcal{G}_{1,2\varepsilon}, \quad \mathcal{G}_{2,2\varepsilon}$ implies that all of the pairs remaining in the end of phase $k^*$ are $\frac{\varepsilon}{W_h(s)}$-good. Thus, $\Delta_h(s, a) \leq \frac{3\varepsilon}{W_h(s)}$.

(3) Assume $(s, a)$ is rejected in phase $k$. By $\mathcal{G}_0$ and Lemma 33,

$$(\text{IV}) < \frac{1}{8}(\bar{\Delta}_{(|\mathcal{Z}_{hi}|A+1-k)} \vee \frac{\varepsilon}{W_h(s)}) \leq \frac{1}{8}(2\Delta_h^{\hat{\pi}}(s, a) \vee \frac{\varepsilon}{W_h(s)}).$$

(i) If $(s, a_s^*)$ is accepted, then it is accepted in phase $k' > k$. Thus,

$$(\text{III}) \leq \frac{1}{8}(\bar{\Delta}_{(|\mathcal{Z}_{hi}|A+1-k')} \vee \frac{\varepsilon}{W_h(s)}) \leq \frac{1}{8}(\bar{\Delta}_{(|\mathcal{Z}_{hi}|A+1-k)} \vee \frac{\varepsilon}{W_h(s)}) \leq \frac{1}{8}(2\Delta_h^{\hat{\pi}}(s, a) \vee \frac{\varepsilon}{W_h(s)}).$$

(ii) If $(s, a_s^*)$ is rejected at phase $k'$, then

$$(\text{III}) < \frac{1}{8}(\bar{\Delta}_{(|\mathcal{Z}_{hi}|A+1-k')} \vee \frac{\varepsilon}{W_h(s)}) \leq \frac{1}{8}(2\Delta_h^{\hat{\pi}}(s, a_s^*) \vee \frac{\varepsilon}{W_h(s)}) \leq \frac{1}{8}(2\Delta_h^{\hat{\pi}}(s, a) \vee \frac{\varepsilon}{W_h(s)})$$

also by $\mathcal{G}_0$ and Lemma 33.

Thus,

$$\Delta_h^{\hat{\pi}}(s, a) - \frac{\varepsilon}{W_h(s)} \leq \Delta_h^{\hat{\pi}}(s, a) - \hat{\Delta}_h^{\hat{\pi}}(s, a) = (\text{II}) \leq (\text{III}) + (\text{IV}) < \frac{1}{4}(2\Delta_h^{\hat{\pi}}(s, a) \vee \frac{\varepsilon}{W_h(s)})$$

If $2\Delta_h^{\hat{\pi}}(s, a) > \frac{\varepsilon}{W_h(s)}$, then $\Delta_h^{\hat{\pi}}(s, a) < \frac{2\varepsilon}{W_h(s)}$ which implies $\Delta_h(s, a) \leq \frac{3\varepsilon}{W_h(s)}$ by the event $\cap_{h'=h+1}^H \mathcal{E}_{h'}$ and Lemma 8.

If $2\Delta_h^{\hat{\pi}}(s, a) \leq \frac{\varepsilon}{W_h(s)}$, then $\Delta_h^{\hat{\pi}}(s, a) < \frac{5\varepsilon}{4W_h(s)}$ which implies $\Delta_h(s, a) \leq \frac{9\varepsilon}{4W_h(s)}$ by the event $\cap_{h'=h+1}^H \mathcal{E}_{h'}$ and Lemma 8.

$\square$

By Lemma 35, we have $|\widehat{\text{OPT}}_h(\varepsilon)| \leq |\text{OPT}_h(3\varepsilon)|$. Now we prove Theorem 3.4.

**Theorem F.1** (Theorem 3.4). *If we run Algorithm 5 with*

$$B \geq \max\{2SHc(\frac{B}{2SH}), 4Hc(\frac{B}{4H}), 2SA\varepsilon_{\frac{B}{2SH}} \log_2 \frac{1}{\varepsilon_{\frac{B}{2SH}}}\}$$

*and an accuracy level $\varepsilon \geq 2SH^2\varepsilon_{\frac{B}{2SH}}$, it uses at most budget $B$ and satisfies the following guarantee:*

$$\mathbb{P}\left(V_0^* - V_0^{\hat{\pi}} > \varepsilon\right) \leq \exp\left(-\tilde{\Theta}\left(\frac{\varepsilon B}{\text{poly}(S, A, H, \log B)}\right)\right)$$

$$+ \exp\left(-\tilde{\Theta}\left(\frac{B}{H^3 \max_{h \in [H]} \sum_{s \in \mathcal{S}} W_h(s)^{-1} \sum_{a \in \mathcal{A}}(\bar{\Delta}_h(s, a) \vee \frac{\varepsilon}{W_h(s)})^{-2}}\right)\right)$$

$$+ \exp\left(-\tilde{\Theta}\left(\frac{\varepsilon^2 B}{H^5 \max_{h \in [H]} |\text{OPT}_h(\varepsilon)|}\right)\right),$$

*where $\text{OPT}_h(\varepsilon) = \{(s, a) \in \mathcal{S} \times \mathcal{A} : \bar{\Delta}_h(s, a)W_h(s) \leq \varepsilon\}$.*

*Proof.* For the budget, the first and the second part each consume at most $\frac{B}{4}$. In the third part total budget of running FB-L2E is at most $\frac{B}{4}$ by Theorem 3.1. We only need to consider the collecting part. Let $K_j = K_j(\delta_j, SAH\delta_j), N_j = \frac{K_j}{2^{j+2}|\widehat{\mathrm{OPT}}_h(\varepsilon)\setminus\cup_{i=1}^{j-1}\mathcal{X}_i|}$. Then, the budget used in collecting is

$$\sum_{j=1}^{J}\frac{K_j}{N_j}n_j = \sum_{j=1}^{J}2^{j+2}|\widehat{\mathrm{OPT}}_h(\varepsilon)\setminus\cup_{i=1}^{j-1}\mathcal{X}_i|\times\frac{B}{H|\widehat{\mathrm{OPT}}_h(\varepsilon)|}2^{-2i-4}\leq\frac{B}{4}$$

Next, we prove the probability bound. Let $\mathcal{M}'_\varepsilon$ be the event where all of the FB-L2E in the third part suceed up to $\varepsilon$ as in Theorem 3.1. Let $\mathcal{L}'_\varepsilon$ be the event where all of the resampling process up to group reachability level $i_\varepsilon$ suceed as in Theorem 3.1. Since $\frac{B}{4H}\leq c(\frac{B}{4H})$,

$$\mathbb{P}(\mathcal{M}'^c_\varepsilon), \mathbb{P}(\mathcal{L}'^c_\varepsilon) \leq \exp\left(-\tilde{\Theta}\left(\frac{\varepsilon B}{\mathrm{poly}(S,A,H,\log B)}\right)\right) \tag{14}$$

by Theorem 3.1. In collecting part, assume the good event

$$\mathcal{H}_{h,\varepsilon} = \{\forall h, \forall s\in\mathcal{Z}_{h,1:i_\varepsilon}, \max_{a'}Q_h^{\hat{\pi}}(s,a')-Q_h^{\hat{\pi}}(s,\hat{\pi}(s))\leq\varepsilon_h(s):=\frac{\varepsilon}{2HJ2^{-j(s)+1}}\},$$

where $i_\varepsilon = \lceil\log_2(\frac{2SH^2}{\varepsilon})\rceil$ and $j(s):=\sup\{j:(s,a')\in\mathcal{X}_j \text{ for some } a'\}$. Let $\tilde{\mathcal{X}}_j=\{s:j(s)=j\}$. Then, $\mathcal{H}_{h,\varepsilon}\implies\mathcal{E}_h$ since

$$\sup_\pi\sum_{s\in\mathcal{Z}_{h1:i_\varepsilon}}w_h^\pi(s)\Delta_h^{\hat{\pi}}(s,\hat{\pi}(s))\leq\frac{\varepsilon}{2HJ}\sup_\pi\sum_{j=1}^{i_\varepsilon}\sum_{s\in\tilde{\mathcal{X}}_j}w_h^\pi(s)2^{-j+1}\leq\frac{i_\varepsilon\varepsilon}{2HJ}\leq\frac{\varepsilon}{2H}.$$

Let $\mathcal{H}_\varepsilon := \cup_{h=1}^H\mathcal{H}_{h,\varepsilon}$, Then the events $\mathcal{M}_\varepsilon, \mathcal{L}_\varepsilon, \mathcal{M}'_\varepsilon, \mathcal{L}'_\varepsilon, \cap_{h=1}^H\mathcal{E}_h, \mathcal{H}_\varepsilon$ and $\mathcal{G}_{0,\varepsilon}(\mathcal{Z}_{hi})$ for all multiple bandit instances $\mathcal{Z}_{hi}$ with $i\leq i_\varepsilon$ implies

$$V_0^* - V_0^{\hat{\pi}} \leq \sum_{h=1}^H\sup_\pi\sum_s w_h^\pi(s)\Delta_h^{\hat{\pi}}(s,\hat{\pi}(s))$$

$$\leq\frac{\varepsilon}{2HJ}\sum_{h=1}^H\sup_\pi\sum_{s\in\mathcal{Z}_{h,1:i_\varepsilon}}w_h^\pi(s)2^{j(s)-1}+H\sum_h\sup_\pi\sum_{s\in\mathcal{Z}_{h,1:i_\varepsilon}^c}w_h^\pi(s)$$

$$\leq\frac{\varepsilon}{2HJ}\sum_{h=1}^H\sup_\pi\sum_{j=1}^{i_\varepsilon}\sum_{s\in\tilde{\mathcal{X}}_j}w_h^\pi(s)2^{j-1}+H\sum_h\sum_{s\in\mathcal{Z}_{h,1:i^*}^c}W_h(s)$$

$$\leq\frac{\varepsilon}{2HJ}\sum_{h=1}^H\sum_{j=1}^{i_\varepsilon}2^{j-1}\sup_\pi\sum_{(s,a)\in\tilde{\mathcal{X}}_j}w_h^\pi(s,a)+H\sum_h|S2^{-i_\varepsilon}$$

$$\leq\frac{\varepsilon}{2HJ}\sum_{h=1}^H\sum_{j=1}^{i_\varepsilon}2^{j-1}\cdot2^{-j+1}+\frac{\varepsilon}{2}$$

$$\leq\frac{\varepsilon}{2}+\frac{\varepsilon}{2}=\varepsilon$$

Thus,

$$\mathbb{P}(V_0^*-V_0^{\hat{\pi}}>\varepsilon)\leq\mathbb{P}(\mathcal{M}_\varepsilon^c\cup\mathcal{L}_\varepsilon^c\cup\mathcal{M}_\varepsilon'^c\cup\mathcal{L}_\varepsilon'^c)+\sum_{h=1}^H\sum_{i=1}^{i_\varepsilon}\mathbb{P}(\mathcal{M}_\varepsilon,\mathcal{L}_\varepsilon,\mathcal{G}_{0,\varepsilon}(\mathcal{Z}_{hi})^c)$$

$$+\sum_{h=1}^H\mathbb{P}(\mathcal{H}_{h,\varepsilon}^c,\mathcal{M}_\varepsilon,\mathcal{L}_\varepsilon,\mathcal{M}_\varepsilon',\mathcal{L}_\varepsilon',\bigcup_{h,i\leq i_\varepsilon}\mathcal{G}_{0,\varepsilon}(\mathcal{Z}_{hi})).$$

The first term is

$$\exp\left(-\tilde{\Theta}\left(\frac{\varepsilon B}{\mathrm{poly}(S,A,H,\log B)}\right)\right)$$

by equation 14 and Theorem 3.1 and the second term is

$$\exp\left(-\tilde{\Theta}\left(\frac{B}{H^3 \max_{h\in[H]} \sum_{s\in\mathcal{S}} W_h(s)^{-1} \sum_{a\in\mathcal{A}} (\bar{\Delta}_h(s,a) \vee \frac{\varepsilon}{W_h(s)})^{-2}}\right)\right)$$

by Lemma 32 and that $i_\varepsilon$ is only a logarithmic factor.

It remains to bound the probability of $\mathcal{H}^c_{h,\varepsilon}$ assuming other events

$$\mathcal{M}_\varepsilon, \mathcal{L}_\varepsilon, \mathcal{M}'_\varepsilon, \mathcal{L}'_\varepsilon, \cap^H_{h'=h+1}\mathcal{E}_{h'}, \bigcup_{h,i\leq i_\varepsilon} \mathcal{G}_{0,\varepsilon}(\mathcal{Z}_{hi}).$$

Let $a^* \in \arg\max_a Q_h^{\hat{\pi}}(s,a)$ and denote $\varepsilon_h(s) := \frac{\varepsilon}{HJ2^{-j(s)+1}}$.

$$\varepsilon_h(s) < \max_a Q_h^{\hat{\pi}}(s,a) - Q^{\hat{\pi}}(s,\hat{\pi}_h(s))$$
$$= Q_h^{\hat{\pi}}(s,a^*) - \hat{Q}_h^{\hat{\pi}}(s,a^*) + \hat{Q}_h^{\hat{\pi}}(s,a^*) - \hat{Q}_h^{\hat{\pi}}(s,\hat{\pi}_h(s)) + \hat{Q}_h^{\hat{\pi}}(s,\hat{\pi}_h(s)) - Q_h^{\hat{\pi}}(s,\hat{\pi}_h(s))$$
$$\leq Q_h^{\hat{\pi}}(s,a^*) - \hat{Q}_h^{\hat{\pi}}(s,a^*) + 0 + \hat{Q}_h^{\hat{\pi}}(s,\hat{\pi}_h(s)) - Q_h^{\hat{\pi}}(s,\hat{\pi}_h(s))$$
$$\implies \left(\varepsilon_h(s)/2 \leq Q_h^{\hat{\pi}}(s,a^*) - \hat{Q}_h^{\hat{\pi}}(s,a^*)\right) \vee \left(\varepsilon_h(s)/2 \leq \hat{Q}_h^{\hat{\pi}}(s,\hat{\pi}_h(s)) - Q_h^{\hat{\pi}}(s,\hat{\pi}_h(s))\right)$$

Then, by Hoeffding's inequality, the probability term below can be obtained as

$$\mathbb{P}(\mathcal{H}^c_{h,\varepsilon} | \mathcal{M}_\varepsilon, \mathcal{L}_\varepsilon, \mathcal{M}'_\varepsilon, \mathcal{L}'_\varepsilon, \cap^H_{h'=h+1}\mathcal{E}_{h'}, \bigcup_{h,i\leq i_\varepsilon} \mathcal{G}_{0,\varepsilon}(\mathcal{Z}_{hi})) \leq$$

$$\mathbb{P}(\exists(s,a) \in \widehat{\mathrm{OPT}}_h(\varepsilon), |\max_{a'} Q_h^{\hat{\pi}}(s,a') - \hat{Q}_h^{\hat{\pi}}(s,a)| \geq \varepsilon_h(s) | \mathcal{M}_\varepsilon, \mathcal{L}_\varepsilon, \mathcal{M}'_\varepsilon, \mathcal{L}'_\varepsilon, \cap^H_{h'=h+1}\mathcal{E}_{h'}, \bigcup_{\mathcal{Z}_{hi}} \mathcal{G}_{0,\varepsilon}(\mathcal{Z}_{hi}))$$

$$\leq \exp\left(-\tilde{\Theta}\left(\frac{1}{2^{2j(s)}} n_0 \cdot \frac{1}{H^2} \cdot \frac{\varepsilon^2}{H^2(2^{-2j(s)})}\right)\right)$$

$$= \exp\left(-\tilde{\Theta}\left(\frac{B}{|\widehat{\mathrm{OPT}}_h(\varepsilon)|} \cdot \frac{\varepsilon^2}{H^5}\right)\right)$$

$$\leq \exp\left(-\tilde{\Theta}\left(\frac{B}{|\mathrm{OPT}_h(3\varepsilon)|} \cdot \frac{\varepsilon^2}{H^5}\right)\right).$$

This proves the theorem. $\qquad\qquad\square$

**Algorithm 5** **B**ackward **R**eachability **E**stimation, **A**ction elimination and **P**olicy refinement (BREAP)