# OpenReview forum: "Instance-Dependent Fixed-Budget Pure Exploration in Reinforcement Learning"
_ICLR.cc/2026/Conference — ICLR 2026 Poster_

### Official Review · Reviewer_81u1 · 2025-10-27

**Soundness:** 3
**Presentation:** 3
**Contribution:** 3
**Rating:** 8
**Confidence:** 3

**Summary:**

The paper introduces the fixed-budget pure-exploration setting for episodic MDPs, where the learner must identify a near-optimal policy using a pre-specified interaction budget without knowing $\delta$ or $\epsilon$ in advance. It proposes BREA (Backward Reachability Estimation and Action-elimination), the first algorithm with fully instance-dependent guarantees in this setting. The authors prove that BREA returns an -optimal policy with probability that decays exponentially in the budget and provides tight, instance-dependent sample-complexity bounds, subsuming multi-armed bandit results.

**Strengths:**

The problem the paper studies is important and the authors present the first algorithm to enjoy instance dependent bounds for such a setting. Being able to design algorithms that do not require knowledge of the confidence parameter $\delta$ or the optimality parameter $\epsilon$  to run the algorithms are more akin to how these algorithms would be used in practice. Furthermore, the author present novel analysis that extends the analysis for the popular SAR algorithm which may be of independent interest to communities who rely on such algorithms to solve problems of interest.

**Weaknesses:**

The main weakness of this papers lies in the fact that the paper considers RL without function approximation and it the computationally feasibility of the proposed algorithm. Specifically whether such ideas would scale to more complicated function approximation schemes or only hold when using the simpler tabular models considered in this work.

**Questions:**

1. While gap dependent bounds are characterization of the hardness of an MDP, there are many MDPs (such as mountain car/cartpole/atari) where the gaps are quite small but we can still get fast rates since we can also characterize the hardness of an MDP by looking at the optimal value function like in [1,2]. Could you also comment on whether we could get bounds that depend on the optimal value function?

2. What are the difficulties in extending these results to setting with function approximation such as linear MDPs [3] or linear mixture MDPs [4].


[1] Zanette, Andrea, and Emma Brunskill. "Tighter problem-dependent regret bounds in reinforcement learning without domain knowledge using value function bounds." International Conference on Machine Learning. PMLR, 2019.

[2] Ayoub, Alex, et al. "Switching the loss reduces the cost in batch reinforcement learning." Forty-first International Conference on Machine Learning. 2024.

[3] Jin, Chi, et al. "Provably efficient reinforcement learning with linear function approximation." Mathematics of Operations Research 48.3 (2023): 1496-1521.

[4] Ayoub, Alex, et al. "Model-based reinforcement learning with value-targeted regression." International Conference on Machine Learning. PMLR, 2020.

---

> ### Author Response · Authors · 2025-11-17
>
> We thank the reviewer for the thoughtful questions.
>
> Answer to Question 1: This is an excellent question. As we briefly noted in the Conclusion, we believe that replacing Hoeffding-type bounds with variance-dependent concentration inequalities could reduce the current $H^2$ factor to a term that depends on the value function or its variance. Such a refinement would move our bounds closer to the value-function–dependent guarantees obtained in, e.g., [1,2]. Developing these sharper bounds requires a more delicate control of the variance of the return under adaptive sampling, and we view this as an interesting direction for future work.
>
> Answer to Question 2: Our action-elimination component removes actions until only one candidate remains in each state. This inherently relies on the action set being finite. That said, in function-approximation settings, one could consider an analogous procedure that maintains a finite set of candidate policies and eliminates them based on confidence intervals—mirroring the spirit of our method. We expect that such an approach could extend some of our ideas, although estimating reachability and value gaps becomes significantly more challenging under function approximation. Exploring this extension is a promising direction for future research.

---

### Official Review · Reviewer_Hors · 2025-10-30

**Soundness:** 3
**Presentation:** 2
**Contribution:** 3
**Rating:** 6
**Confidence:** 3

**Summary:**

The paper proposes a novel algorithm called BREA for reward-free exploration of Markov decision processes in the fixed-budget setting. Concretely, the authors show that the probability of the proposed algorithm of failing to achieve epsilon-accuracy is inversely proportional to the suboptimality gaps and epsilon.

**Strengths:**

To the best of my knowledge, the fixed-budget analysis of the proposed exploration algorithm is novel and correct, though I did not carefully check the proofs.

**Weaknesses:**

My main concern about the paper is conceptual. Fixed-budget algorithms usually prove a result on the accuracy or regret at the end of learning, and the authors also do this. However, the authors also prove results on sample complexity, which seems backwards when the budget is fixed beforehand. To compute a budget based on the sample complexity, one would need prior knowledge of several problem-specific parameters, which goes against the contribution of needing only B as input. Several theorems state as a *result* that the total budget is at most B, which seems like a tautology.

Several theoretical results are difficult to interpret since they include parameters hidden inside functions. It would be useful to include a qualitative comparison with the sample complexity of algorithms in the fixed-confidence setting.

Though a minor point, I do not agree with some of the claimed benefits of the fixed-budget setting. Concretely, the fixed-confidence setting has a clearly defined stopping rule and the aim is to guarantee PAC-compliance of the returned policy, while analyzing the precise sample complexity. Hence the algorithm cannot use as many samples as possible (the number of samples is bounded by the sample complexity), and the quality of the returned policy is guaranteed by the PAC-condition.

**Questions:**

In the expression for \epsilon_B on page 4, what is c(B)?

The idea of rewarding specific state-action pairs is similar to active coverage (Al Majrani et al., COLT 2023), did you explore this connection?

In Remark 4 the sample complexity \tau depends on C_L2E(B/(2SH)), which in turn depends on B. Should not \tau be independent of B?

---

> ### Author Response · Authors · 2025-11-17
>
> We thank the reviewer for the thoughtful questions.
>
> Answer to Question 1:
> The term $c(B)$ is $\tilde{O}(\mathrm{poly}(S, A, H))$ and is defined in Appendix C, lines 1130–1132. We chose to hide this term because the first exponential term in Theorem 3.3 is a lower-order term with respect to $\epsilon$, and we view the second exponential term as the main term of the bound. Including the explicit—yet cumbersome—expression of $c(B)$ in the theorem would unnecessarily obscure the presentation for readers, especially since it does not affect the main takeaway. A similar notational choice is also used in [1].
> That said, as the reviewer correctly points out, a first-time reader of Theorem 3.3 may naturally wonder what $c(B)$ represents. Therefore, if the paper is accepted, we will consider adding the explicit expression of $c(B)$ directly to the statement of the theorem.
>
> Answer to Question 2: We appreciate the reviewer for raising the connection with [2]. Although both our algorithm and COVGAME (their proposed method) employ reward functions that promote visiting informative state–action pairs, the two approaches differ fundamentally in both objective and mechanism.
>
> First, active coverage aims to exactly satisfy a prescribed set of sampling requirements $c(h,s,a)$ by solving a min–max coverage game. The reward functions in COVGAME are adversarially generated weights $\lambda_t$ over $(h,s,a)$, designed to approximate the optimal coverage distribution that minimizes the active-coverage complexity $\phi^\*(c)$. In contrast, our backward reachability estimation step is not designed to solve a coverage game; instead, following [1], it resets the reward to a binary indicator to estimate the reachability $W_h(s)$ of each state. This estimation is only used to calibrate the instance-dependent accuracy thresholds required for the action-elimination analysis.
>
> Second, the role of the reward signal is conceptually different. In COVGAME, the varying reward $\lambda_t$ is the core mechanism that drives the learner toward a globally optimal coverage distribution. In our algorithm, the reward modification serves a local objective: for each state $s$ and step $h$, the reward is chosen so that an optimal policy for the modified MDP maximizes the visitation probability of $(s,1)$. This allows us to reuse the collected data across all states for reachability estimation under the fixed-budget constraint, which is specific to our $\epsilon$-uniform guarantee analysis in the fixed-budget setting.
>
> Finally, we note that active coverage is designed for the PAC setting and assumes that the sampling process stops only after coverage constraints are met. In contrast, our algorithm must operate under a strict episode budget $B$ and derive tail guarantees on the resulting simple regret — a fundamentally different objective. Because of this, the complexity measure $\phi^\*(c)$ and its game-theoretic machinery are incompatible with the fixed-budget requirement of our setting.
>
> While the conceptual similarity in “rewarding” certain state–action pairs is appreciated, the underlying goals and theoretical tools differ substantially, and our algorithm does not directly fit into the active coverage framework. Nevertheless, the reviewer’s suggestion is valuable, and we will consider adding a discussion of this conceptual relation.
>
> Answer to Question 3: This is a valid point. We will revise the expression to use the polynomial notation, consistent with the notation in Remark 6. This simplification is possible because $C_{\mathrm{L2E}}(B / 2SH)$ grows only logarithmically with respect to $B$.
>
> Reference
>
> [1] A. Wagenmaker et al., “Beyond No Regret: Instance-dependent PAC Reinforcement Learning”, COLT 2022.
>
> [2] A. Al-Marjani et al., “Active Coverage for PAC Reinforcement Learning”, COLT 2023.

---

### Official Review · Reviewer_Fd9H · 2025-10-31

**Soundness:** 3
**Presentation:** 3
**Contribution:** 3
**Rating:** 6
**Confidence:** 3

**Summary:**

This paper studies the problem of fixed-budget pure exploration in finite-horizon episodic Markov Decision Processes (MDPs). The objective is to identify a near-optimal policy given a predetermined, fixed budget of interactions, which contrasts with the more common fixed-confidence (PAC) setting where the error level $\epsilon$ and failure rate $\delta$ are provided as inputs.

The authors propose a new algorithm, BREA, which operates by working backward from the final step $H$. The algorithm integrates two main components: A fixed-budget reward-free exploration mechanism and an action elimination phase. The authors establish an instance-dependent, $\epsilon$-uniform guarantee for this fixed-budget setting in MDPs. This guarantee upper bounds the probability of returning a $>\epsilon$ suboptimal policy for all $\epsilon$ above a budget-dependent threshold. As byproducts, the paper also provides $\epsilon$-uniform guarantees for fixed-budget reward-free exploration and for the SAR multiple-bandit algorithm.

**Strengths:**

1. The paper addresses the fixed-budget pure exploration problem in MDPs, a practical setting that has received little attention in RL theory.
2. The introduction provides a clear argument for the merits of the fixed-budget setting over the standard fixed-confidence setting.
3. The paper introduces the $\epsilon$-uniform guarantee, a new analytical approach for this problem. The new analysis for the SAR algorithm and the fixed-budget reward-free exploration mechanism are also solid contributions.
4. The BREA algorithm's design, which separates reachability estimation from action elimination, is logical and well-explained.

**Weaknesses:**

1. The most significant weakness is the $H^5$ factor in the sample complexity of the main result in Theorem 3.3. This is substantially worse than the $H^4$ dependence often seen in related PAC RL literature. The authors acknowledge this, but it remains a major limitation of the paper's primary result.
2. The paper introduces a variant algorithm in Section 3.4 that achieves a more comparable sample complexity to prior work (like MOCA). However, this theorem is stated informally, and the algorithm details are deferred to the appendix. This makes the paper's strongest result less clear and accessible.
3. The paper is highly technical, which makes it challenging to parse. Key components and definitions being in the appendix add to this difficulty.

**Questions:**

1.  Could the authors elaborate on the $H^5$ dependence in Theorem 3.3? The paper attributes this to the algorithm's lack of foresight in distributing the budget across steps $h$. Is this $H^5$ term an artifact of the BREA algorithm's two-phase design, or do you believe it is a fundamental lower bound for any fixed-budget algorithm in this setting?
2.  The variant algorithm in Section 3.4 seems to be the one that yields a sample complexity in Remark 6 most comparable to the MOCA algorithm. Why is this presented as a "variant" rather than the main result? If a user does have a target accuracy $\epsilon$ in mind, should they use this variant over Algorithm 3?
3.  The $\epsilon$-uniform guarantee for the SAR algorithm (Theorem 3.2) is presented as a contribution of independent interest. How does the resulting bound compare to existing non-$\epsilon$-uniform fixed-budget guarantees for the multiple-bandit problem?

---

> ### Author Response · Authors · 2025-11-17
>
> We thank the reviewer for the thoughtful questions.
>
> Answer to Question 1:
> As we noted in the Conclusion, we believe that the $H^2$ factor may be improved by leveraging variance-dependent concentration inequalities. However, we conjecture that an $H^3$ dependence is unavoidable for algorithms that learn sequentially across the horizon, and this sequential dependency inherently accumulates across the $H$ stages.
>
> Answer to Question 2:
> This is an excellent point. In the paper, we place strong emphasis on the $\epsilon$-uniform guarantee. From this perspective, the variant algorithm receives a theoretical guarantee only for a single fixed value of $\epsilon$. For this reason, we chose to highlight BREA rather than the variant. If the paper is accepted, the camera-ready version allows one additional page, and we will consider including this variant algorithm in the main text.
>
> Answer to Question 3:
> For the multiple-bandit problem, existing results provide guarantees only for the case $\epsilon = 0$ [1]. Our results match the order of that guarantee when $\epsilon = 0$, and therefore can be viewed as a generalization of the previously known $\epsilon = 0$ result.
>
> Reference
>
> [1] S. Bubeck et al., Multiple Identifications in Multi-Armed Bandits, ICML 2013.

---

### Official Review · Reviewer_HDZ3 · 2025-10-31

**Soundness:** 3
**Presentation:** 3
**Contribution:** 3
**Rating:** 8
**Confidence:** 4

**Summary:**

The paper studies fixed-budget pure exploration in reinforcement learning, where the goal is to identify a near-optimal policy using a limited number of environment interactions. The authors propose a new algorithm that achieves ε-uniform guarantee given budget B (clearly ε=ε(B)).  They also analyze the SAR algorithm in multi-bandit problems.

**Strengths:**

From practical perspective, budget is more meaningful and easier to set than epsilon. As a result, the authors study 'the right problem.'
The assumptions made are reasonable and aligned with similar works.

I did not check the proofs, but the statements are clear and show 'what is promised.'

**Weaknesses:**

Some notation is weak, i.e., has loose ends.

The usual weakness for such a paper is the cliche: there are no experiments. I don't take this against the authors. I think no experiments is fine.

The MAB part comes across as an unnecessary material.

**Questions:**

Definition of W_h(s) in line 139: there is dangling a. I assume it is meant \pi(s) instead of a.
What is S in 148? Has it been defined earlier? I wasn't able to find it.
In line 186, why does epsilon_h(s) satisfy the conditions in Proposition 1?
In Theorem 3.1, who to interpret B>=c(B)? Does it mean: there exists \hat{B} such that for every B>=\hat{B}, ...

---

> ### Author Response · Authors · 2025-11-17
>
> We thank the reviewer for the thoughtful questions.
>
> Answer to Question 1: In line 139, we intentionally wrote $W_h(s)=\sup_\pi w_h^\pi(s)=\sup_\pi w_h^\pi (s,a)$ to indicate that introducing a separate notation for $W_h(s,a)$ is unnecessary. For any state-action pair $(s,a)$, we can consider a policy that takes action $a$ at state $s$ at step $h$, so the expression remains consistent.
>
> Answer to Question 2: The symbol $S$ in line 148 denotes the size of the state space $\mathcal{S}$, which is defined in line 84-85.
>
> Answer to Question 3: In line 186, we sketch why the desired inequality (lines 188–198) holds when Proposition 1 is satisfied with appropriate values of \epsilon_h(s). Our main analysis then shows that Proposition 1 indeed holds with these values of \epsilon_h(s) with high probability.
>
> Answer to Question 4: In Theorem 3.1, our intention is to state that Algorithm 1 is run with a budget $B$ satisfying $B\geq c(B)$. That is, there exists a value $\hat{B}$ such that the theorem applies for all $B\geq \hat{B}$.

---

### Meta-Review · Area_Chair_LyYs · 2026-01-03

**Summary:**

This paper studies fixed-budget pure exploration in episodic MDPs and makes a strong theoretical contribution by introducing BREA, the first algorithm with fully instance-dependent, ε-uniform guarantees in this setting, together with novel analytical tools for reward-free exploration and multiple bandits. Reviewers consistently agreed that the paper addresses an important and practically relevant problem, that the guarantees are technically sound, and that the ε-uniform perspective is both novel and valuable; the extension of SAR to ε-uniform fixed-budget guarantees was also viewed as an independent contribution. The main concerns focused on the suboptimal dependence on the horizon in the primary bound, the relegation of a stronger variant algorithm to a secondary role, and some conceptual and presentation issues around interpreting sample-complexity statements under a fixed budget. The authors’ rebuttal satisfactorily clarified notation, justified their design and presentation choices, and acknowledged the remaining limitations as directions for future work rather than flaws. Overall, given the consistency of positive evaluations, the novelty of the guarantees, and the solid technical execution, I recommend acceptance.

**Reviewer Scores:**

NA

---

### Decision · Program_Chairs · 2026-01-26

Accept (Poster)